energy/civil engineering

energy evolution, tension bolts, UDEC, differential analysis, rock

**Author for correspondence:**
Xiangyu Wang
e-mail: wangxiangyu_cumt@163.com

†Present address: Nanhu campus, China University of Mining and Technology, No.1, University Road, Xuzhou City, Jiangsu Province, China.

# An investigation on the effect of high energy storage anchor on surrounding rock conditions

Bowen Wu[1,2], Xiangyu Wang[1,2,†], Jianbiao Bai[2],
Wenda Wu[1,2], Ningkang Meng[1,2] and Huasheng Lin[3]

[1]School of Mines, China University of Mining & Technology, and [2]State Key Laboratory of Coal Resources and Safe Mining, Xuzhou 221116, People's Republic of China
[3]School of Minerals and Energy Resources Engineering, University of New South Wales, Sydney, NSW, 2052, Australia

BW, 0000-0003-1652-2391; XW, 0000-0002-6148-2598;
JB, 0000-0002-1369-9792; WW, 0000-0001-7373-4353;
NM, 0000-0002-0670-0214; HL, 0000-0001-5646-8495

High pre-tension bolt is an effective strata control technique and is the key to ensure the stability of anchorage and roadway. Based on the performances of high energy storage tension rock bolts in different rock properties, this study proposed a constitutive model to describe the energy balance of anchor under uniaxial compression. UDEC was used to simulate the behaviour of anchor in coal under uniaxial compression and the results were analysed to study the rock mechanical properties, degree of damage and energy evolution. Simulation results showed that tension rock bolts can improve the mechanical properties and energy storage capacities of the anchor. The energy evolution was divided into three stages: (i) the external work was stored in the form of elastic strain energy ($U^e$) in the anchor prior to the yielding strength; (ii) the elastic strain energy reached its maximum near the peak strength; (iii) energy was dissipated from fracture friction ($W_f$), plastic deformation ($W_p$) and acoustic emission ($U^r$) during post-peak stage. The installation of tension rock bolts was more suitable for medium hard rock (e.g. sandy mudstone), whereas it was not effective for hard rock (e.g. sandstone).

## 1. Introduction

Rock strength is a critical factor for underground structural design. To improve the mechanical properties of rock, rock and cable bolts are generally implemented. In underground coal mines, the rock strengthening measure is immediate bolt support after excavation to ensure the roadway stability and subsequent workplace safety [1]. Extensive practical work revealed that this is an effective strata control method, which has also been widely used in Chinese coal mines [2].

To determine the effect of rock bolt on anchor strengthening and improve the ability of anchorage, a significant body of researches has been carried out. Hou & Gou [3] theoretically analysed the effect of rock bolts on the peak and the residual strengths of rock around the anchorage range. Based on the improvement of mechanical properties prior to and after the peak strength, Hou & Gou [3] proposed a constitutive model for rock strengthening effect around the roadway with rock bolts. Kang et al. [4] described the mechanisms and technologies of rock bolt supporting system in China. The authors suggested that high strength-stiffness-reliability and low-density proactive rock bolts can improve the strength of surrounding rock. Wang et al. [5] derived the equations of shear strength prior to and after anchorage. The analysis showed that once the grouting is completed, the shear strength of anchorage range was improved together with the mechanical properties of surrounding rock (elastic modulus, cohesion, friction angle and dilation angle). Wei & Gou [6] carried out physical simulation tests to study the influence of pre-tension on anchor strength and deformation and found that high pre-tension can improve the anchorage strength. Based on the triaxial physical simulation tests, Wang et al. [7] revealed that the rock bolt does not only improve the capacity of fracture surrounding rock, but also controls the discontinuous deformation of the fractured surrounding rock. This will in turn improve the continuity and integrity of surrounding rock mass of deep roadway.

In the numerical simulation, the bolt section in anchorage range can be represented by the material model as well as structural elements. Both methods have been proven successful in simulating the mechanical properties of bolts and anchors under various loading conditional. Meng [8] imported the constitutive model of anchor in the fractured rock to FLAC3D and subsequently simulated the effect of anchoring condition on ultimate bearing capacity. Results showed that the higher pre-tension and density can improve the ultimate bearing capacity. Zuo et al. [9] used elastic-isotropic model in FLAC3D to study the influence of the alignment between bolt and borehole diameters on anchorage effect. Numerical simulation suggested a reasonable alignment (4–11 mm diameter difference) can improve the anchorage effect. Ho et al. [10] used elasto-plastic model to study the fracture mechanism of the interface between rock bolt and anchoring agent under various horizontal stress conditions. Thereby, Grasselli [11], Aziz & Jalalifar [12] and Tatone et al. [13] also used material model to simulate various laboratory condition of rock bolt. However, the mesh size is an important factor which can influence the simulation results using the material model. This constrains the preciseness of the simulation. On the other hand, the structural element approach can overcome this problem. At present, the structural element can be divided into 'cable element' and 'rock-bolt element'. Zhang et al. [14] used cable element to study the parameters including length, anchorage length and rock anchor spacing via FLAC3D and proposed several suggestions to support bolt design. Wu et al. [15] attempted to simulate the supporting element (e.g. wooden cylinder) in the supported roadway using cable element and found that the rock bolts can improve the capacity of wooden cylinder and prevent the fracture development. Pull-test or underground excavation with rock-bolt support were simulated by Vardakos et al. [16], Malmgren & Nordlund [17], Li et al. [18], Gao et al. [19], Shreedharan & Kulatilake [20]. Ma et al. [21] used rock-bolt element to simulate pull-test via FLAC2D and studied the relationship between rock bolt and rock as well as the mechanical properties of newly developed rock bolt and its preferable conditions.

According to the previous studies, it can be found that the focus was mainly on the anchorage technique, rock bolt, coordination effect and stress transfer. Although there is an overall agreement that high pre-tension can improve the mechanical properties of anchor, the influence of high pre-tension under various rock strength conditions on anchorage was not investigated. Majority of the research analysed the anchorage effect via elasto-plastic model and stress–strain relationship, while there were limited studies investigating the problem using the principle of energy conservation. Importantly, the fracture evolution of anchorage is a process of energy dissipation and release. Hence, it is more reasonable to study the anchorage effect using energy law [22]. Many numerical simulations use cable element to approach this topic; this is inappropriate considering the method is one-dimensional element which consists of two degrees of freedom. On the other hand, rock-bolt element is more suitable for the analysis as it is a two-dimensional element with three degrees of freedom [23]. The paper used rock-bolt element through UDEC Trigon to study the influence of high pre-tension on anchor with various surrounding rock strengths under uniaxial compression. This paper aims to study the different strengthening behaviours of tension rock bolt under various conditions based on the monitor of the energy evolution, the stress–strain relation as well as the fracture development in different rock strengths.

## 2. Energy balance and components

Nowadays, the study of energy evolution in rock mechanics has been refined to different depth of cover, loading conditions and in situ stress magnitudes [24–28]. Energy changes determined in UDEC are

performed for the intact rock, the joints and for the work done on boundaries. Without considering dynamic calculation, the energy balance in UDEC mainly involves work done on boundary, stored strain energy and dissipated energy [23]. This study determined the constitutive law based on the previous studies as well as the simulation conditions.

In uniaxial compression, work done from the external force is denoted by $W$, and $U^e$ is used to represent the elastic strain energy due to elastic deformation of the anchor. The difference in work done between boundary and elastic strain energy is dissipated energy, $U^d$. Hence, total input energy can be written as follows:

$$W = U^d + U^e. \tag{2.1}$$

Energy is mainly dissipated in three ways. First, energy is dissipated by friction due to the fracture development in rock; this is denoted as $W_f$. Second source is due to the plastic deformation of rock ($W_p$). Once the rock has undergone plastic deformation, energy is dissipated from plastic work done. The rest of energy is dissipated via acoustic emission, denoted as $U^r$. Therefore, $U^d$ can be expressed as follows:

$$U^d = W_f + W_p + U^r. \tag{2.2}$$

In UDEC, the incremental change of these energy components is cumulative and determined by time steps [23]. By combining equations (2.1) and (2.2), $W$ can be expressed as follows:

$$W = W_f + W_p + U^r + U^e. \tag{2.3}$$

Subsequently, this energy balance equation can be used to investigate the complex energy dissipation process in an anchor.

# 3. Parameters calibration

## 3.1. The UDEC Trigon approach

UDEC Trigon model was proposed by Gao *et al.* [29], aiming at simulating brittle fracture of rock. In this model, a rock is represented by an assembly of triangular blocks bonded together via the grain contacts. Each block is made elastically by dividing them into triangular finite-difference zones. In the direction normal to a contact, the stress-displacement relation is assumed to be linear and governed by the stiffness $k_n$, such that

$$\Delta\sigma_n = -k_n\Delta u_n, \tag{3.1}$$

where $\Delta\sigma_n$ is the effective normal stress increment and $\Delta u_n$ is the normal displacement increment. A limiting tensile strength ($T$) is assumed for the contact. If this value is exceeded, then $\sigma_n = 0$.

In the shear direction, the response is governed by constant shear stiffness. The shear stress, $\tau_s$, is determined by a combination of contact properties, cohesion ($c$) and friction angle ($\varphi$), where

$$|\tau_s| \leq c + \sigma_n\tan\varphi = \tau_{\max}, \tag{3.2}$$

then

$$\Delta\tau_s = -k_s\Delta u_s^e, \tag{3.3}$$

however, if $|\tau_s| \geq \tau_{\max}$, then

$$\tau_s = \text{sign}(\Delta u_s^e)\tau_{\max}, \tag{3.4}$$

where $\Delta u_s^e$ is the elastic component of the incremental shear displacement and $\Delta u_s$ is the total incremental shear displacement.

The proposed modelling approach has been implemented in UDEC [23].

## 3.2. Mechanical parameters of coal and rock mass

Since blocks in UDEC Trigon model are considered as elastic materials, they cannot be plastically destroyed. However, the anchor breakage is plastic damage. To closely mimic the mechanical properties and energy evolution of anchor post-peak strength, this study also used strain-softening model in UDEC. The strain-softening model is based on the UDEC Mohr–Coulomb model with non-associated shear and associated tension flow rules.

**Table 1.** Intact rock properties and scaled rock mass properties of coal measures from Yuwu coal mine.

| | intact rock | | | rock mass | | |
|---|---|---|---|---|---|---|
| lithology | $E_r$ (GPa) | $\sigma_r$ (MPa) | RQD (%) | $E_m$ (GPa) | $\sigma_{cm}$ (MPa) | $\sigma_{tm}$ (MPa) |
| coal | 2.6 | 10.8 | 75 | 0.79 | 5.1 | 0.51 |
| sandy mudstone | 5.4 | 35.5 | 90 | 3.1 | 25 | 2.5 |
| sandstone | 9.7 | 53.5 | 88 | 5.2 | 36.1 | 3.6 |

**Table 2.** Calibrated mechanical parameters of blocks and joints of the coal measures.

| | density (kg m$^{-3}$) | Young's modulus (GPa) | cohesion (MPa) | friction (°) | tensile strength (MPa) |
|---|---|---|---|---|---|
| block | 1400 | 0.79 | 1.6 ($\varepsilon_p = 0$) | 27 | 0.9 |
| | | | 1.1 ($\varepsilon_p = 0.04$) | | |
| | | | 0.6 ($\varepsilon_p = 0.15$) | | |
| | normal stiffness (GPa) | tangential stiffness (GPa) | cohesion (MPa) | friction (°) | tensile strength (MPa) |
| joint | 113 | 45.2 | 1.3 | 18 | 0.4 |

**Table 3.** Calibrated mechanical parameters of blocks and joints of sandy mudstone.

| | density (kg m$^{-3}$) | Young's modulus (GPa) | cohesion (MPa) | friction (°) | tensile strength (MPa) |
|---|---|---|---|---|---|
| block | 1800 | 3.1 | 8.0 ($\varepsilon_p = 0$) | 27 | 2.5 |
| | | | 5.0 ($\varepsilon_p = 0.06$) | | |
| | | | 2.0 ($\varepsilon_p = 0.15$) | | |
| | normal stiffness (GPa) | tangential stiffness (GPa) | cohesion (MPa) | friction (°) | tensile strength (MPa) |
| Joint | 372 | 149 | 6.2 | 18 | 1.8 |

This study selected coal (soft), sandy mudstone (medium) and sandstone (strong) to represent various rock strengths. The intact properties of the rock mass are listed in table 1. These properties were obtained through laboratory compression tests and were provided by the Yuwu coal mine. The rock-quality designation (RQD) values of the rock mass were evaluated from borehole televiewer images.

The rock mass elastic modulus was calculated using the relationship between RQD and the elastic modulus ratio [30], as shown in equation (3.5), where $E_m$ is the elastic modulus of the rock mass and $E_r$ is the elastic modulus of the intact rock sample.

$$\frac{E_m}{E_r} = 10^{0.0186\text{RQD}-1.91}. \tag{3.5}$$

The rock mass strength was then calculated using the relation between the unconfined compressive strength ratio $\sigma_{cm}/\sigma_c$ and the deformation modulus ratio $E_m/E_r$ [31]. The value of $q$ is 0.63 [32].

$$\frac{\sigma_{cm}}{\sigma_c} = \left(\frac{E_m}{E_r}\right)^q. \tag{3.6}$$

To represent the coal and rock by using an assembly of triangular blocks, the properties of the blocks and contacts were calibrated against the coal measure and rock mass properties listed in tables 2–4. This was achieved by simulating unconfined compression tests in a numerical model created using the Trigon logic. The size of the rock sample is 1 m (in width) × 2 m (in height) [31] (figure 1). The bottom of the

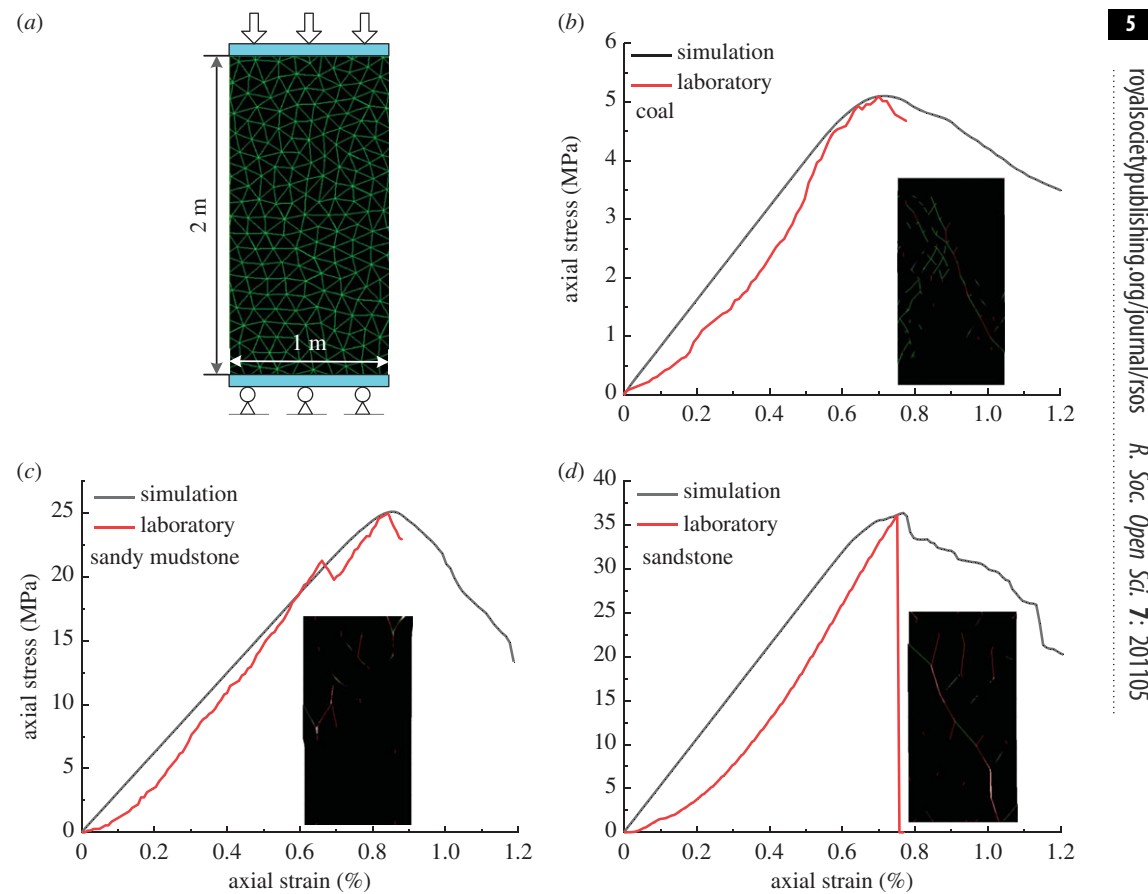

**Figure 1.** Results of uniaxial compressive strength tests (*a*) numerical simulation of uniaxial compression, (*b*) stress–strain curve of coal, (*c*) stress–strain curve of sandy mudstone, (*d*) stress–strain curve of sandstone.

**Table 4** Calibrated mechanical parameters of blocks and joints of sandstone.

|  | density (kg m$^{-3}$) | Young's modulus (GPa) | cohesion (MPa) | friction (°) | tensile strength (MPa) |
|---|---|---|---|---|---|
| block | 2550 | 5.2 | 13.0 ($\varepsilon_p = 0$) | 30 | 2.5 |
|  |  |  | 7.0 ($\varepsilon_p = 0.04$) |  |  |
|  |  |  | 4.0 ($\varepsilon_p = 0.10$) |  |  |

|  | normal stiffness (GPa) | tangential stiffness (GPa) | cohesion (MPa) | friction (°) | tensile strength (MPa) |
|---|---|---|---|---|---|
| joint | 665 | 266 | 11.2 | 18 | 1.8 |

numerical model was fixed and a displacement rate of 0.02 m s$^{-1}$ was applied at the top. The calibrated properties of the UDEC model are illustrated in table 5.

The uniaxial compressive strength and elastic modulus data derived by numerical simulation are similar to the data obtained from laboratory tests (within an error of 9%). Hence, the micromechanical parameters of the coal mass and the rock mass were properly calibrated.

### 3.3. Rock-bolt element parameter

In UDEC, rock-bolt element is different from cable element. Rock-bolt element is a two-dimensional element in which its two nodes have three degrees of freedom (two displacements and one rotation). It can resist bending and can yield along the axial direction. Figure 2 illustrates the components of

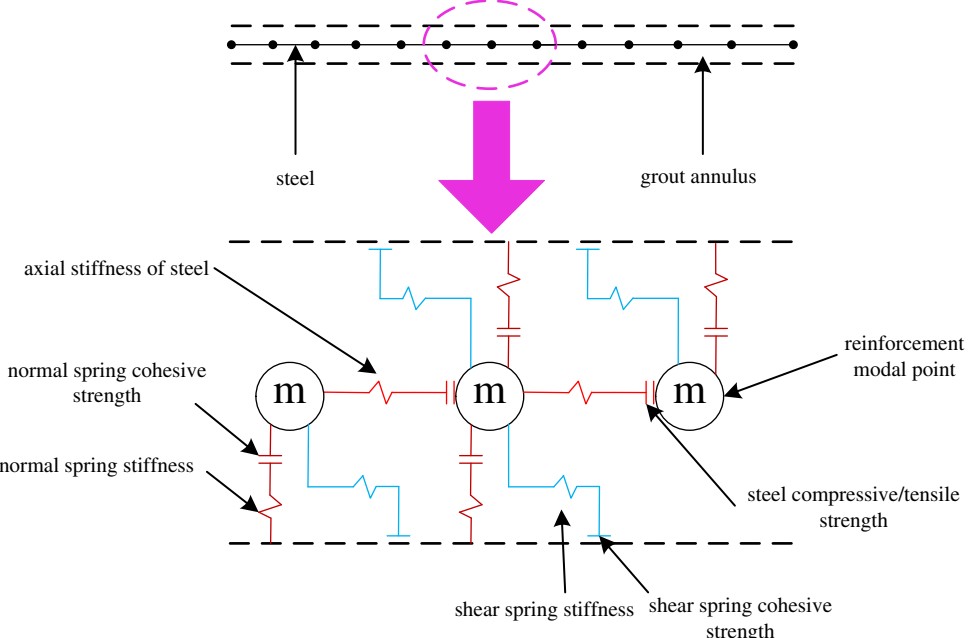

**Figure 2.** Conceptual mechanical representation of the rock-bolt element.

**Table 5.** Calibrated microproperties in the UDEC Trigon model to represent the rock mass.

| lithology | Young's modulus (GPa) | | | compressive strength (MPa) | | |
|---|---|---|---|---|---|---|
| | target | calibrated | error (%) | target | calibrated | error (%) |
| coal | 0.79 | 0.73 | 7 | 5.1 | 5.1 | 0 |
| sandy mudstone | 3.1 | 2.9 | 6 | 25 | 25.1 | 0.4 |
| sandstone | 5.2 | 4.7 | 9 | 36.1 | 36.3 | 5 |

rock-bolt element. Rock bolts interact with UDEC via shear and normal coupling springs, which are nonlinear connectors. These are used to transfer force and motion between rock-bolt element and the mesh nodes. Figure 2 shows material behaviour of shear and normal coupling spring for rock-bolt elements. Shear coupling springs are used to simulate the shear behaviour of anchorage length, and normal coupling springs are used to simulate the compression of surrounding rock.

Rock-bolt element can be broken at the node. The bolt breakage can be simulated by defining the tensile failure strain limit (*tfstrain*). Rock bolt will be deemed as failed if $\varepsilon_{pl} \geq tfstrain$, where *tfstrain* has to be defined. $\varepsilon_{pl}$ is the total tensile yield strain of any element of anchor

$$\varepsilon_{pl} = \sum \varepsilon_{pl}^{ax} + \sum \frac{d}{2} \frac{\theta_{pl}}{L}, \tag{3.7}$$

where $\varepsilon_{pl}^{ax}$ is the axial deformation, d is rock bolt diameter and $\theta_{pl}$ is the average rotation angle of the component.

This study used rock-bolt element to simulate HRB335 threaded steel anchor at 22 mm diameter. Based on the laboratory pull-test results, the input parameters of rock-bolt element can be obtained, as displayed in table 6.

# 4. Research background and model establishment

The common way to apply pre-tension is to install the nut on the tail thread of rock bolt. By applying torque to the nut, axial tension is acting onto the rock bolt [33]. However, high pre-tension is difficult to control in practice. For instance, when the torque achieves 300–400 N m, the pre-tension can sometimes still be low. This low conversion efficiency is due to the torque at a thread. To increase the pre-tension of rock bolt, a new type of anchor lock has been developed by the research team. This

**Table 6.** Rock bolt and anchor parameters.

| | cross-section area (m$^2$) | elastic modulus (GPa) | tensile yield strength (kN) | second moment of area (m$^4$) | tfstrain |
|---|---|---|---|---|---|
| rock bolt | $3.8 \times 10^{-4}$ | 200 | 198 | $1.2 \times 10^{-8}$ | 0.012 |
| | exposed perimeter (m) | cohesive strength of shear coupling spring (MPa) | stiffness of shear coupling spring (GPa) | frictional resistance of the shear coupling spring (°) | cohesive strength of normal coupling spring (MPa) | stiffness of normal coupling spring (GPa) | frictional resistance of the normal coupling spring (°) |
| anchor parameters | 0.07 | 1 | 8 | 45 | 200 | 20 | 0 |

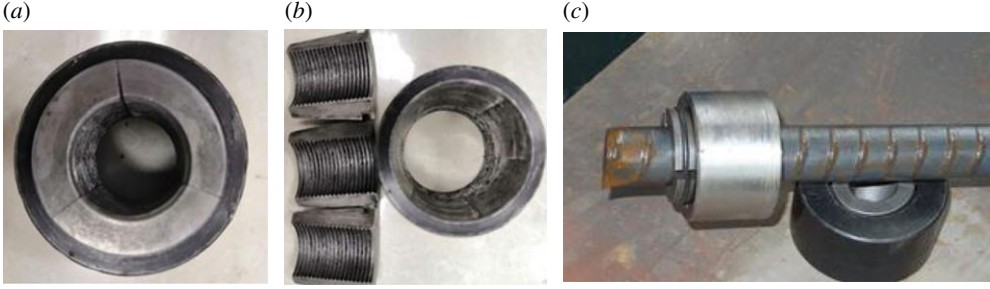

**Figure 3.** Rock-bolt barrel and wedge and assembly drawing.

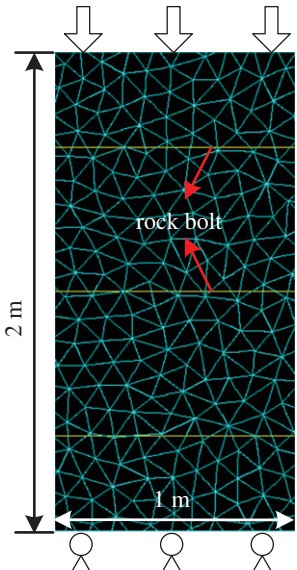

**Figure 4.** Model overview and dimensions.

**Table 7.** Numerical simulation scheme.

|  | rock-bolt existence | pre-tension (load) | displacement rate m s$^{-1}$ |
|---|---|---|---|
| coal (soft) | no | / | 0.02 |
|  | yes | 0.365 MPa | 0.02 |
| sandy mudstone (medium) | no | / | 0.02 |
|  | yes | 0.365 MPa | 0.02 |
| sandstone (strong) | no | / | 0.02 |
|  | yes | 0.365 MPa | 0.02 |

development changed the traditional pre-tension method, which is capable of applying high pre-tension without damaging rock bolt. The anchor lock is composed of an anchor ring and three clips. The anchor ring is a ring-shaped structure, in which the centre hole is tapered. The clips are disposed between the inner wall of the tapered hole of the anchor ring and the anchor body, as displayed in figure 3.

Figure 4 shows the set-up of UDEC Trigon simulation. The blocks and interfaces used strain-softening and Coulomb slip models, respectively. The dimensions of the numerical model are $1 \times 2$ m. Due to the limitation of rock-bolt element, the effect of vertically installed rock bolts was not significant. Hence, rock bolts were installed horizontally to the rock body at $y = 0.4$, 1.0 and 1.6 m. Pre-tension was applied by applying load at the nodes of the rock bolts. In practice, the common torque applied to the rock bolt is approximately 300–400 N m, which are 40–50 kN pre-tension correspondingly [34]. To achieve high pre-tension in the numerical model, the study set the pre-tension as 70% of the yielding strength, i.e.

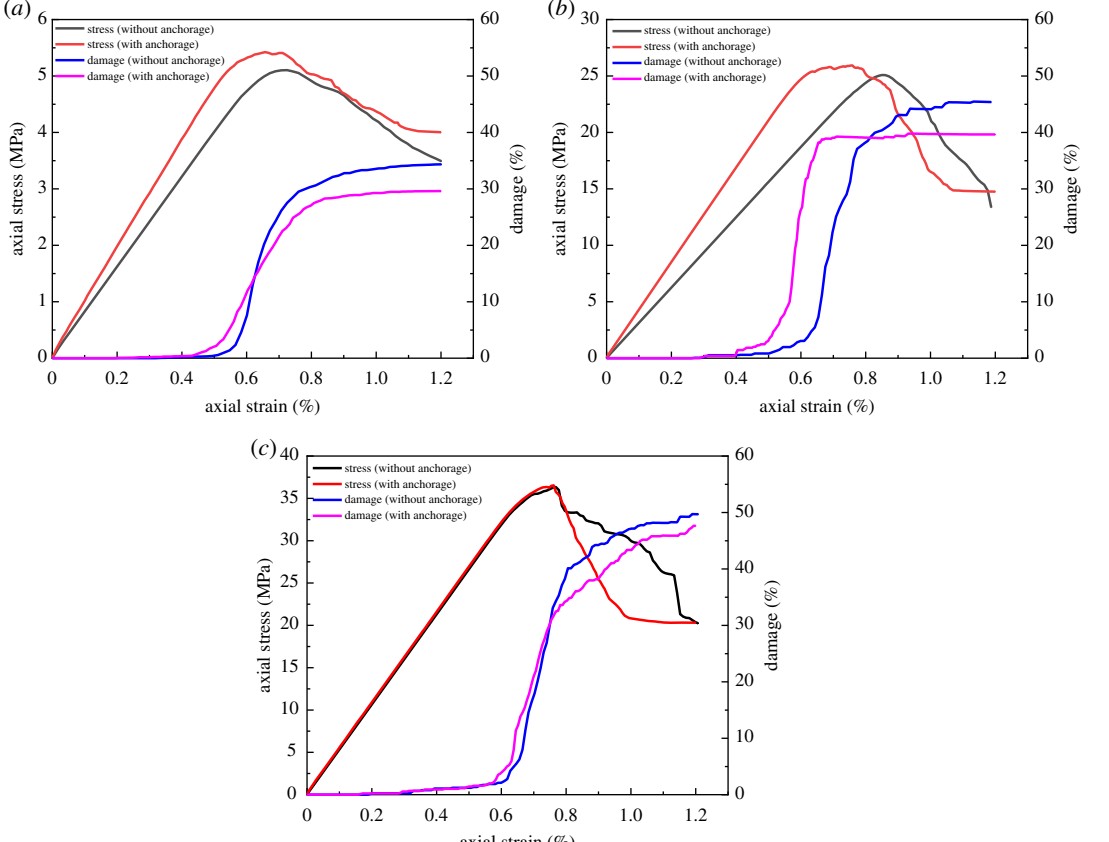

**Figure 5.** The stress and degree of damage of rock with and without anchorage (*a*) coal; (*b*) sandy mudstone; (*c*) sandstone.

138.6 kN/0.365 MPa. The displacement rate of 0.02 m s$^{-1}$ was applied at the top boundary of the model while fixing the bottom boundary. During the simulation process, the stress–strain relationship and fracture development of anchor were monitored by FISH function. Boundary, frictional and plastic work were recorded using UDEC energy element. 'SET energy on' command was used to activate the energy monitoring function while turn off 'mass-scaling'. As this study used non-viscous boundary which does not consider the dynamic calculation, damping was set as 'auto'. The simulation procedures involved: (i) model construction and calibration, and (ii) test simulation. Simulation plan as shown in table 7.

# 5. Discussion

## 5.1. Mechanical properties and damage analysis

To monitor the degree of damage of rock and anchor, the total fracture length as well as the shear and tensile fracture lengths during uniaxial compression were measured. The anchor and surrounding rock fracture are a complex process, in which the shear and tensile failures can occur in a single fracture. Hence, only the initial (first) failure type was recorded for each fracture. The degree of damage (*D*) can be calculated as follows [35]:

$$D = \frac{L_S + L_T}{L_C} \times 100\%, \tag{5.1}$$

where $L_C$ is the total fracture length, $L_S$ and $L_T$ are total shear and tensile fracture lengths, respectively.

Figure 5 illustrates the stress and degree of damage of rock during the process with and without anchorage. Based on the comparison, the mechanical properties (peak strength, residual strength and elastic modulus) of rock were increased with the installation of tension rock bolts, while the degree of damage decreased. Under the same axial strain, the higher strength always led to a higher degree of damage regardless of the anchorage condition. The installation of tension rock bolts did not have a significant influence on the axial strain of rock during damage. There was an increase in the degree of

**Table 8.** Mechanical properties and degree of damage of rock prior to and after anchorage.

| | rock bolt existence | elastic modulus (GPa) | peak strength (MPa) | residual strength (MPa) | degree of damage (%) |
|---|---|---|---|---|---|
| coal (soft) | no | 0.73 | 5.1 | 3.8 | 34.3 |
| | yes | 0.76 | 5.4 | 4.0 | 29.6 |
| sandy mudstone | no | 2.9 | 25.1 | 13 | 45.3 |
| (medium) | yes | 3.5 | 26 | 14.7 | 39.6 |
| sandstone | no | 4.7 | 36.3 | 20.2 | 49.7 |
| (strong) | yes | 4.8 | 36.5 | 20.3 | 47.6 |

damage prior to the peak rock strength and the gradient became flattered when it was near the peak strength. To compare the effect of tension rock bolts under different rock conditions, table 8 summarizes the critical parameters from figure 5.

Table 8 suggested that there are differences in the strengthening effect of tension rock bolts on rock mechanical properties with various rock types. When the bolt was installed in coal, elastic modulus, peak strength and residual strength increased 4.1% (0.03 GPa), 5.9% (0.3 MPa) and 5.2% (0.2 MPa), while the degree of damage decreased 13.7%. In sandy mudstone, elastic modulus, peak strength and residual strength increased 20.7% (0.6 GPa), 3.6% (0.9 MPa) and 13.1% (1.7 MPa), while the degree of damage decreased 12.6%. For sandstone, elastic modulus, peak strength and residual strength increased 2.1% (0.1 GPa), 0.6% (0.2 MPa) and 0.5% (0.1 MPa), while the degree of damage decreased 4.2%. Sandy mudstone had the most change in absolute magnitudes and sandstone had the least change. In terms of the relative magnitude difference, coal had the most change whereas sandstone had the least. The absolute magnitude change of sandy mudstone was the most while its degree of damage was only in the middle, indicating the mechanical properties of rock were improved by the installation of rock bolts. However, this is not the primary reason for the reduction in degree of damage, in which the main reason was due to the prevention of fracture development and coalescence using rock bolts. Sandstone had a low change in terms of absolute and relative magnitudes. This suggests that high pre-tension rock bolts do not have significant influence on strong rock. This is because the rock strength was high and it required substantially higher stress to induce the rock failure. Under this condition, the effect of rock bolt is negligible comparing with the high strength. At post-peak of the loading process, the fracture was more likely to be developed due to the low plasticity, which was difficult to control using rock bolts.

## 5.2. Analysis of the law of energy evolution

During the loading process, rock was continuously deforming and storing elastic strain energy. At the same time, the energy was dissipated due to internal damage. When the elastic strain energy reached the limit, it was released and led to rock failure. Hence, the failure process of rock is closely related to energy evolution. Based on the analysis carried out in §5.1, tension rock bolts are not suitable for hard rock. Therefore, this section will focus on the energy evolution of coal and sandy mudstone, as displayed in figures 6 and 7.

In figures 6 and 7, the energy dissipation due to acoustic emission ($U^r$) was illustrated solely. This is because the energy dissipation curve of acoustic emission was similar to the accumulative acoustic emission, which is a major indication of anchor breakage. Second, the energy dissipation due to acoustic emission was substantially lower than the other sources. A sole illustration can represent the energy evolution of acoustic emission more clearly. According to the results from figures 6 and 7, the energy evolution of rock was consistent regardless of the rock bolts. Based on the correspondence between the stress curve and energy curve, the energy evolution of rock can be divided into three stages:

Stage 1: from the start of the load to the yielding strength. External work was fully absorbed by rock and stored in the form of elastic strain energy. The energy curve increased nonlinearly during this stage, while the external work curve was well aligned with the elastic strain curve. Based on the stress and

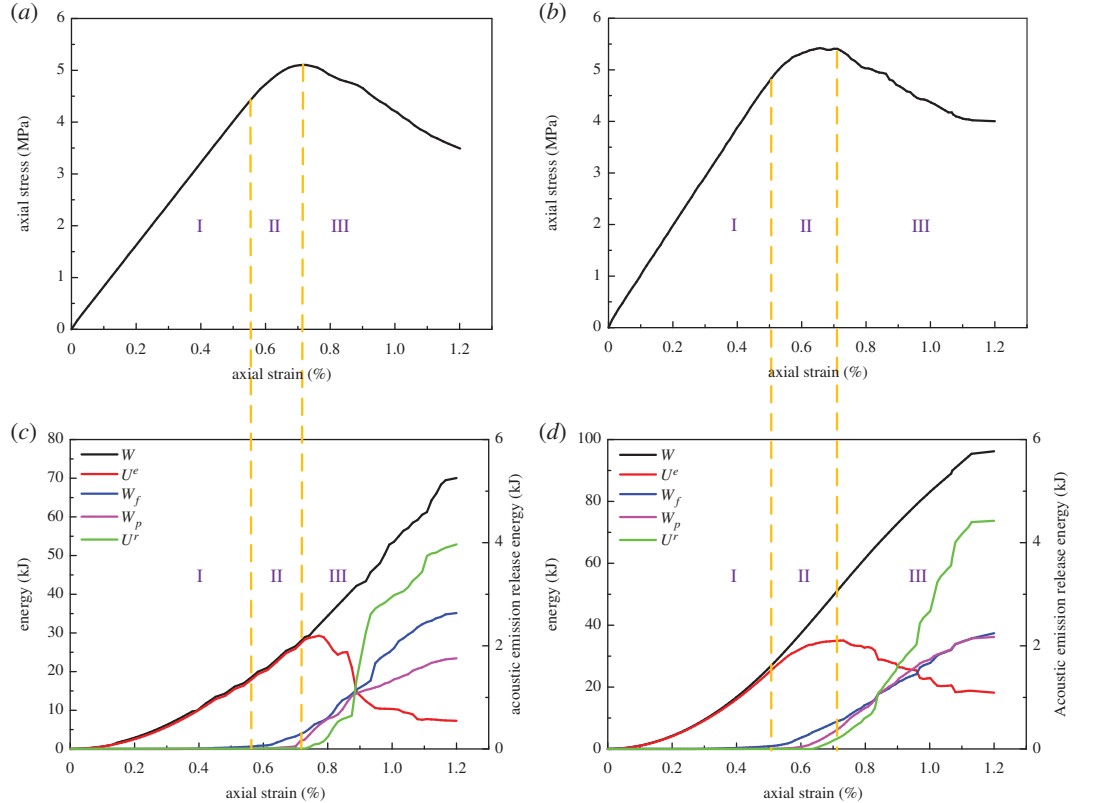

**Figure 6.** Stress and energy evolution with and without anchorage in coal (*a*) stress–strain curve of coal (without anchorage); (*b*) stress–strain curve of coal (with anchorage); (*c*) energy evolution of coal (without anchorage); (*d*) energy evolution of coal (with anchorage).

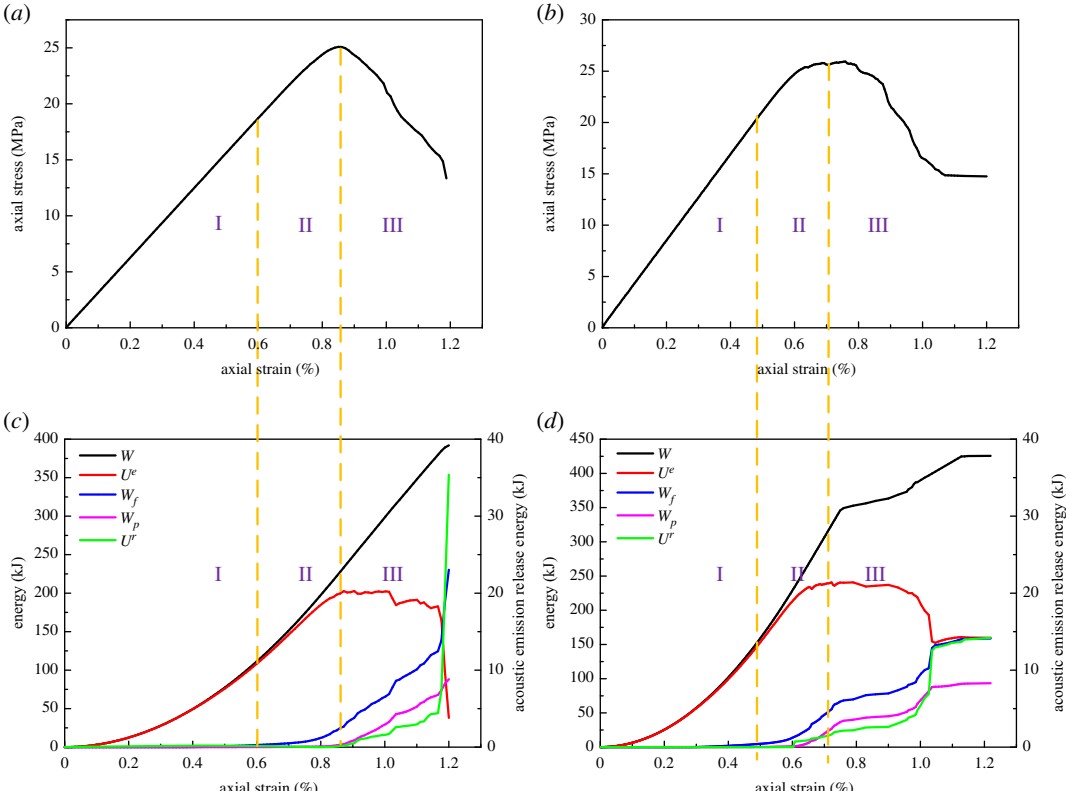

**Figure 7.** Stress and energy evolution with and without anchorage in sandy mudstone (*a*) stress–strain curve of sandy mudstone (without anchorage); (*b*) a stress–strain curve of sandy mudstone (with anchorage); (*c*) energy evolution of sandy mudstone (without anchorage); (*d*) energy evolution of sandy mudstone (with anchorage).

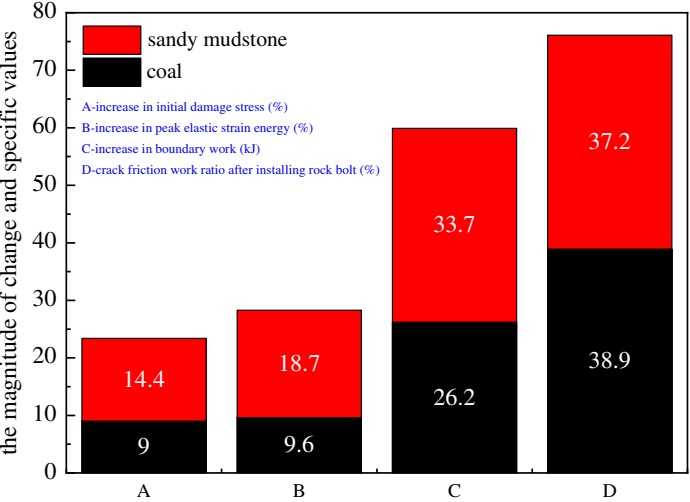

**Figure 8.** The comparison between coal and sandy mudstone on different energy categories.

energy curves, the rock was under elastic deformation prior to the yielding strength. There was no observable internal damage, such that the internal energy dissipation was considered as none.

Stage 2: from the yielding strength to the peak strength. External work continuously increased and the total energy that the rock absorbed was also increasing. However, as the rock was in plastic condition and there was some degree of internal damage, the frictional work in-between cracks and plastic deformation started to occur and increased at a high rate. This indicates that the degree of damage was increased although the acoustic emission level was still low. During this stage, the elastic strain energy was also increasing but the growth rate was flatter. Thereby, its alignment with the external work was getting worse and reached its maximum near the peak strength point.

Stage 3: post-peak stage. External work continuously increased but the elastic energy in the rock was released at a high rate. There was a considerable increase in the energy dissipation due to frictional work, plastic deformation and acoustic emission. The acoustic emission energy curve was growing rapidly, suggesting that the degree of damage was high. The external work was mainly transferred to the energy required for rock fracture towards the end of the simulation and the elastic strain energy was not stored any more. When the stress dropped to the residual strength, the elastic strain energy and energy dissipation curves became flatter, although the energy continuously dissipated.

Although the energy evolution of rock was consistent regardless of the existence of rock bolts, the degree of damage and other parameters changed noticeably with the installation of rock bolts. First, there were increasing in stress and decreasing in the strain at anchor damage with rock bolts. For coal, the stress increased from 4.4 to 4.8 MPa and the strain reduced from 0.57% to 0.5%. For sandy mudstone, stress increased from 18 to 20.6 MPa and strain reduced from 0.6% to 0.49%. Second, the peak elastic energy increased from 29.2 to 35 kJ and from 202.7 to 240.7 kJ for coal and sandy mudstone, respectively. When strain reached 1.2%, the external work of coal and sandy mudstone increased 26.2 and 33.7 kJ. For coal, the work done from fracture friction was 35.1 kJ without rock bolt, accounting for 50.1% of the total input energy, whereas the energy was 37.4 kJ and took account of 38.9% with rock bolt. On the other hand, for sandy mudstone, the work done from fracture friction was 230.3 kJ without rock bolt, accounting for 58.8% of the total input energy, whereas the energy was 158.7 kJ and took account of 37.2% with rock bolt. Based on aforementioned data, the installation of tension rock bolts can improve the strength of surrounding rock and its storage of elastic strain energy. At the same time, it can also prevent fracture development and postpone the elastic strain energy release at post-peak. Figure 8 shows the comparison of the energy categories. From the figure, it can be seen that the performance of sandy mudstone was better than coal. Hence, the indication here is that the tension rock bolts can improve the mechanical properties and energy storage capacity of rock, which agrees with the conclusion from §5.1. Thereby, the most suitable rock type for the application is medium hard rock, e.g. sandy mudstone.

Figure 9 depicts the stress and energy evolution of sandstone with and without anchorage. According to the figure, it can be found that the relationships of sandstone follow the same trend as those of coal. The difference between sandstone and coal was that the changes of frictional work and plastic

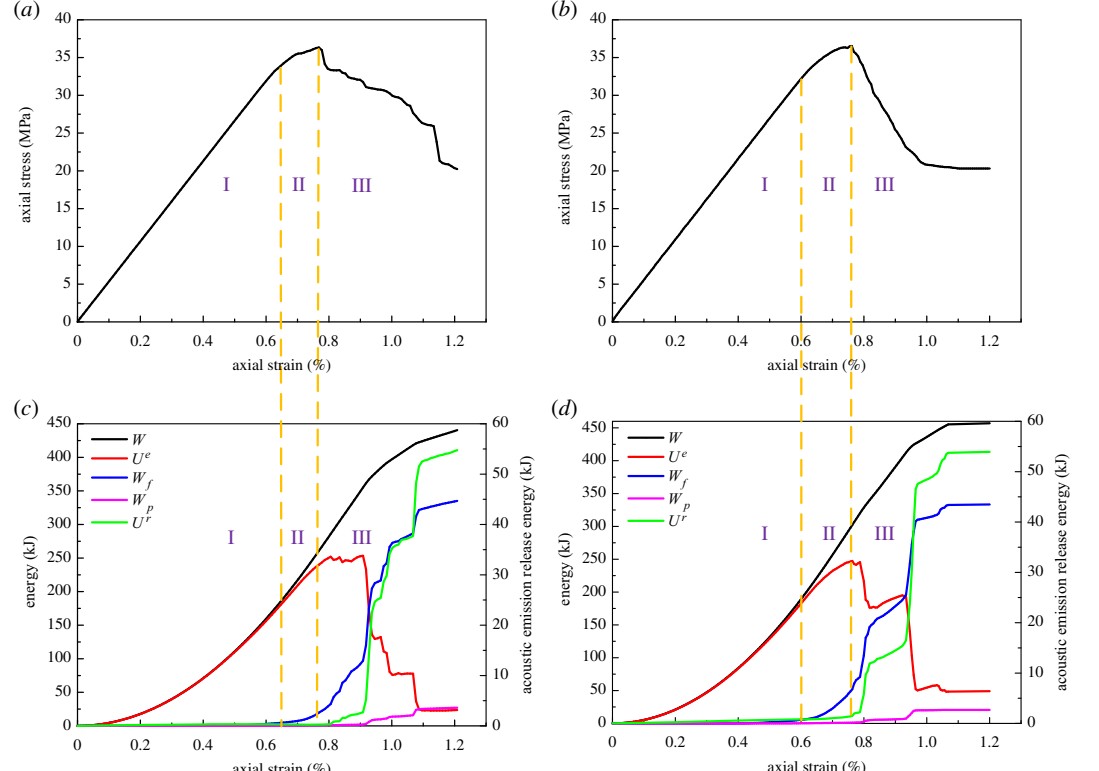

**Figure 9.** The stress and energy evolution of sandstone with and without anchorage (*a*) stress–strain relationship curve of sandstone (without anchorage); (*b*) stress–strain relationship curve of sandstone (with anchorage); (*c*) energy evolution of sandstone (without anchorage); (*d*) energy evolution of sandstone (with anchorage).

deformation of sandstone were insignificant with rock bolts, indicating that the installation of rock bolts does not increase the plasticity of sandstone nor prevent fracture development. The plastic deformation curve was always at a low level while the frictional work curve was always at high level. This phenomenon again supports that fracture development was the main reason of rock failure.

# 6. Conclusion

Based on the UDEC simulation on the strengthening effect of tension rock bolts in different rock properties, a number of parameters were studied, including mechanical properties, degree of damage and energy evolution. The following conclusions were drawn from the numerical results:

(1) Tension rock bolts can improve the mechanical properties and energy storage capacity of anchor. Subsequently, it can reduce the degree of damage of the anchor while increasing the elastic strain energy storage and plastic deformation work. By suppressing energy dissipation due to fracture friction and delaying the post-peak elastic strain release, the stability of anchor can be increased.

(2) Based on the stress–strain and energy dissipation–strain relationships during uniaxial compression, the energy evolution of anchor was divided into three stages. In the first stage, the external work was fully absorbed by rock and was stored as elastic deformation energy. In the second stage, the axial stress surpasses the rock yielding strength, which work was done from fracture friction and plastic deformation. At the same time, the growth rate of elastic strain energy was reduced and the elastic strain energy reached the maximum value near the peak strength point. In the third stage, the energy stored in the rock was rapidly released from elastic strain energy. The work from fracture friction, plastic deformation and acoustic emission increased substantially. At the end of the process, the energy of external work was mainly transferred to rock failure, where the energy was dissipated via fracture friction, plastic deformation and acoustic emission. Among them, work from fracture friction and plastic deformation was the major contributing factor for energy dissipation.

(3) Based on stress and energy evolution analysis on different rock types, it can be seen that the improvement of mechanical properties of rock due to rock bolts was the secondary factor for the degree of damage reduction, whereas its effect on preventing fracture development and coalescence was the main contributing factor. Tension rock bolts are more suitable for medium hard rock (e.g. sandy mudstone), followed by soft rock (e.g. coal). The tension rock bolts were not effective for hard rock. The implementation could not prevent fracture development within the rock mass. The application effect is weak in hard rock (e.g. sandstone).

Data accessibility. Our data are deposited at Dryad Digital Repository: https://doi.org/10.5061/dryad.cvdncjt1w [36].
Authors' contributions. B.W. and X.W. conceived the article structure. B.W. and W.W. performed the rock mechanical experiments. J.B. developed energy balance criterion. B.W. numerical simulation. All the authors analysed the data. N.M. prepared the plots. H.L. language revision. All the authors prepared the initial draft and revised the paper.
Competing interests. We have no competing interests.
Funding. This work is supported by the Fundamental Research Funds for the Central Universities (grant no. 2019BSCX05).
Acknowledgements. The authors are grateful for the valuable comments from the editors and the reviewer, which has substantially improved the quality of our work.

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
