## [Reviewer comments · Royal Society Open Science]

Review History

RSOS-201105.R0 (Original submission)

Review form: Reviewer 1 (Rudrajit Mitra)

Is the manuscript scientifically sound in its present form?

Yes

Are the interpretations and conclusions justified by the results?

Yes

Is the language acceptable?

No

Do you have any ethical concerns with this paper?

No

Have you any concerns about statistical analyses in this paper?

No

Recommendation?

Major revision is needed (please make suggestions in comments)

Comments to the Author(s)

Good study, however, it needs to have some major changes as specified in the annotated paper (Appendix A). Some of the points are not clear in the paper, which needs to be clarified.

Review form: Reviewer 2**Is the manuscript scientifically sound in its present form?**

Yes

Are the interpretations and conclusions justified by the results?

Yes

Is the language acceptable?

Yes

Do you have any ethical concerns with this paper?

No

Have you any concerns about statistical analyses in this paper?

No

Recommendation?

Major revision is needed (please make suggestions in comments)

Comments to the Author(s)

Comments to the Editor - Authors

This article investigates the influence of high pre-tension on anchor with various surrounding rock strengths under uniaxial compression, aiming to study the different strengthening behaviors of tension rock bolt under various conditions based on monitor of the energy evolution, the stress-strain relation as well as the fracture development in different rock strengths.

The subject of the study is interesting as the results provide a scientific basis for the rational stability control of coal roadway and an improvement in safety of coal mines.

It is necessary however to make minor and major revisions before publication of the article.

For the minor revisions, please find attached (Appendix B) the reviewed manuscript with several corrections and comments.

Regarding the major revisions:

6.1 Mechanical properties and damage analysis

Table 8 presents the degree of damage for each rock type, with and without anchorage, at a particular strain level of 1.2%. It is not clear to the reader the reasons that the authors chose this level for damage degree comparison and not for example that one corresponds to the peak stress or any other strain level.

Please explain and clarify.

6.2 Analysis of the law of energy evolution

Page 7, lines 28-29:

Fig. 5 shows that the degree of damage increasing in Stage 1 (before the yielding strength). Please comment and clarify on this.

Page 7, lines 34-35:

This is in contrast to Fig. 5. Fig. 5 shows that the degree of damage is increasing at Stage 2. Please comment and clarify on this.

7 Conclusions

In this section there are some points that are mentioned extensively more than one time in Discussion section. For example, the point in page 8, lines 42-43 is repeated in page 7, lines 1-2 and 9-10 as well as in page 8, lines 16-17.

The authors should take care and reform properly the Discussion and the Conclusion section.

Regards

Decision letter (RSOS-201105.R0)

Dear Dr Wang

The Editors assigned to your paper RSOS-201105 "An Investigation on the effect of high energy storage anchor on surrounding rock conditions" have now received comments from reviewers and would like you to revise the paper in accordance with the reviewer comments and any comments from the Editors. Please note this decision does not guarantee eventual acceptance.

Please submit your revised manuscript and required files (see below) no later than 21 days from today's (ie 10-Aug-2020) date. Note: the ScholarOne system will 'lock' if submission of the revision is attempted 21 or more days after the deadline. If you do not think you will be able to meet this deadline please contact the editorial office immediately.

Kind regards,
Royal Society Open Science Editorial Office
Royal Society Open Science

on behalf of Professor Zach Agioutantis (Associate Editor) and R. Kerry Rowe (Subject Editor)
openscience@royalsociety.org

Associate Editor Comments to Author (Professor Zach Agioutantis):

Associate Editor: 1

Comments to the Author:

Please document all changes to the original document.

Reviewer comments to Author:

Reviewer: 1

Comments to the Author(s)

Good study, however, it needs to have some major changes as specified in the annotated paper. Some of the points are not clear in the paper, which needs to be clarified.

Reviewer: 2

Comments to the Author(s)

Comments to the Editor - Authors

This article investigates the influence of high pre-tension on anchor with various surrounding rock strengths under uniaxial compression, aiming to study the different strengthening behaviors of tension rock bolt under various conditions based on monitor of the energy evolution, the stress-strain relation as well as the fracture development in different rock strengths.

The subject of the study is interesting as the results provide a scientific basis for the rational stability control of coal roadway and an improvement in safety of coal mines.

It is necessary however to make minor and major revisions before publication of the article.

For the minor revisions, please find attached the reviewed manuscript with several corrections and comments.

Regarding the major revisions:

6.1 Mechanical properties and damage analysis

Table 8 presents the degree of damage for each rock type, with and without anchorage, at a particular strain level of 1.2%. It is not clear to the reader the reasons that the authors chose this level for damage degree comparison and not for example that one corresponds to the peak stress or any other strain level.

Please explain and clarify.

6.2 Analysis of the law of energy evolution

Page 7, lines 28-29:

Fig. 5 shows that the degree of damage increasing in Stage 1 (before the yielding strength). Please comment and clarify on this.

Page 7, lines 34-35:

This is in contrast to Fig. 5. Fig. 5 shows that the degree of damage is increasing at Stage 2. Please comment and clarify on this.

7 Conclusions

In this section there are some points that are mentioned extensively more than one time in Discussion section. For example, the point in page 8, lines 42-43 is repeated in page 7, lines 1-2 and 9-10 as well as in page 8, lines 16-17.

The authors should take care and reform properly the Discussion and the Conclusion section.

Regards

===PREPARING YOUR MANUSCRIPT===

===PREPARING YOUR REVISION IN SCHOLARONE===

-- If you have uploaded ESM files, please ensure you follow the guidance at <https://royalsociety.org/journals/authors/author-guidelines/#supplementary-material> to include a suitable title and informative caption. An example of appropriate titling and captioning may be found at https://figshare.com/articles/Table_S2_from_Is_there_a_trade-off_between_peak_performance_and_performance_breadth_across_temperatures_for_aerobic_sc_ope_in_teleost_fishes_/3843624.

Author's Response to Decision Letter for (RSOS-201105.R0)

See Appendix C.

Decision letter (RSOS-201105.R1)

Dear Dr Wang

On behalf of the Editors, we are pleased to inform you that your Manuscript RSOS-201105.R1 "An Investigation on the effect of high energy storage anchor on surrounding rock conditions" has been accepted for publication in Royal Society Open Science subject to minor revision in

accordance with the referees' reports. Please find the referees' comments along with any feedback from the Editors below my signature.

Please submit your revised manuscript and required files (see below) no later than 7 days from today's (ie 11-Sep-2020) date. Note: the ScholarOne system will 'lock' if submission of the revision is attempted 7 or more days after the deadline. If you do not think you will be able to meet this deadline please contact the editorial office immediately.

Finally, please note that the email address 'baijanbiao@cumt.edu.cn' is currently marked as invalid in our system. Please check and adjust this when resubmitting.

on behalf of Professor Zach Agioutantis (Associate Editor) and R. Kerry Rowe (Subject Editor)
openscience@royalsociety.org

Associate Editor Comments to Author (Professor Zach Agioutantis):

The authors have revised the document according to the reviewers comments. However, there are still a number of typos and odd expressions in the document

- a) please correct "targe" in Table 5
- b) Is "tension" in Tables 2,3,4 "tensile strength"?
- c) rock mass does not have plural form. Please correct rock masses to rock mass
- d) the sentence "The installation of tension rock bolts was more suitable for medium hard rock (e.g. sandy mudstone), whereas it was not effective for hard rock (e.g. sandstone)" is repeated in the summary text twice.
- e) and fracture development of an anchor was monitored by FISH function. -> why are you talking about fractures of the anchor?
- f) What is a Coulomb-sliding model. That is not a standard term.

===PREPARING YOUR MANUSCRIPT===

===PREPARING YOUR REVISION IN SCHOLARONE===

- If you are providing image files for potential cover images, please upload these at this step, and inform the editorial office you have done so. You must hold the copyright to any image provided.
- A copy of your point-by-point response to referees and Editors. This will expedite the preparation of your proof.

- Ensure that your data access statement meets the requirements at <https://royalsociety.org/journals/authors/author-guidelines/#data>. You should ensure that you cite the dataset in your reference list. If you have deposited data etc in the Dryad repository, please only include the 'For publication' link at this stage. You should remove the 'For review' link.
- If you are requesting an article processing charge waiver, you must select the relevant waiver option (if requesting a discretionary waiver, the form should have been uploaded at Step 3 'File upload' above).
- If you have uploaded ESM files, please ensure you follow the guidance at <https://royalsociety.org/journals/authors/author-guidelines/#supplementary-material> to include a suitable title and informative caption. An example of appropriate titling and captioning may be found at https://figshare.com/articles/Table_S2_from_Is_there_a_trade-off_between_peak_performance_and_performance_breadth_across_temperatures_for_aerobic_scope_in_teleost_fishes_/3843624.

Author's Response to Decision Letter for (RSOS-201105.R1)

See Appendix D.

Decision letter (RSOS-201105.R2)

Dear Dr Wang,

It is a pleasure to accept your manuscript entitled "An Investigation on the effect of high energy storage anchor on surrounding rock conditions" in its current form for publication in Royal Society Open Science.

on behalf of Professor Zach Agioutantis (Associate Editor) and R. Kerry Rowe (Subject Editor)
openscience@royalsociety.org

Appendix A**ROYAL SOCIETY
OPEN SCIENCE****An Investigation on the effect of high energy storage
anchor on surrounding rock conditions**

Journal:	Royal Society Open Science
Manuscript ID	RSOS-201105
Article Type:	Research
Date Submitted by the Author:	26-Jun-2020
Complete List of Authors:	Wu, Bowen; China University of Mining and Technology; State Key Laboratory of Coal Resources and Safe Mining Wang, Xiang-yu; China University of Mining and Technology; State Key Laboratory of Coal Resources and Safe Mining Bai, Jianbiao; State Key Laboratory of Coal Resources and Safe Mining Wu, Wenda; China University of Mining and Technology Meng, Ningkang; China University of Mining and Technology, School of Mines Lin, Huasheng; University of New South Wales
Subject:	Energy < ENGINEERING AND TECHNOLOGY, Civil engineering < ENGINEERING AND TECHNOLOGY
Keywords:	Energy evolution, tension bolts, UDEC, differential analysis, rock
Subject Category:	Engineering

Author-supplied statements

Relevant information will appear here if provided.

Ethics

Does your article include research that required ethical approval or permits?:

This article does not present research with ethical considerations

Statement (if applicable):

CUST_IF_YES_ETHICS :No data available.

Data

It is a condition of publication that data, code and materials supporting your paper are made publicly available. Does your paper present new data?:

Yes

Statement (if applicable):

Our data are deposited at Dryad.

Review URL: <https://datadryad.org/stash/share/Agf-g6O9idvPjujUn9zqr8m79qBwHC6MF69F5n4fWk8>

doi:10.5061/dryad.cvdncjt1w

Conflict of interest

I/We declare we have no competing interests

Statement (if applicable):

We have no competing interests.

Authors' contributions

This paper has multiple authors and our individual contributions were as below

Statement (if applicable):

B.W. and X.W. conceived the article structure. B.W. and W.W. performed the rock mechanical experiments. J.B. developed energy balance criterion. B.W. numerical simulation. All the authors analysed the data. N.M. prepared the plots. H.L. language revision. All the authors prepared the initial draft and revised the paper.

An Investigation on the effect of high energy storage anchor on surrounding rock conditions

Bowen Wu ^{1,2}, Xiangyu Wang ^{1,2*}, Jianbiao Bai ², Wenda Wu ^{1,2},
Ningkang Meng ^{1,2}, Huasheng Lin ³

¹ School of Mines, China University of Mining & Technology, Xuzhou 221116, China

² State Key Laboratory of Coal Resources and Safe Mining, Xuzhou 221116, China

³ School of Minerals and Energy Resources Engineering, University of New South Wales, Sydney, NSW, 2052, Australia

Keywords: Energy evolution, tension bolts, UDEC, differential analysis, rock

1. Summary

High pre-tension bolt is an effective strata control technique and is the key to ensure the stability of anchorage and roadway. Based on the performances of high energy storage tension rock bolts in different rock properties, this study proposed a constitutive model to describe the energy balance of anchor under uniaxial compression. UDEC was used to simulate the behaviour of anchor in coal under uniaxial compression and *the* results were analysed to study the rock mechanical properties, degree of damage and energy evolution. Simulation results showed that tension rock bolts can improve the mechanical properties and energy storage capacities of anchor. The energy evolution was divided into three stages: i) the external work was stored in the form of elastic strain energy (U^e) in the anchor prior to the yielding strength; ii) the elastic strain energy reached its maximum near the peak strength; iii) Energy was dissipated from fracture friction (W_f), plastic deformation (W_p) and acoustic emission (U^r) during post-peak stage. The installation of tension rock bolts resulted in the most improvement of surrounding rock condition for medium hard rock including sandy mudstone, whereas it had the least impact on hard rock, e.g. sandstone.

2. Introduction

Rock strength is a critical factor for underground structural design. To improve the mechanical properties of rock, rock and cable bolts are generally implemented. In underground coal mines, the rock strengthening measure is immediate bolt support after excavation to ensure the roadway stability and subsequent workplace safety [1]. Extensive practical work revealed that this is an effective strata control method, which has also been widely used in Chinese coal mines [2].

To determine the effect of rock *been* bolt on anchor strengthening and improve the ability of anchorage, a significant body of researches have carried out. Hou [3] theoretically analysed the effect of rock bolts on the peak and the residual strengths of rock around anchorage range. Based on the improvement of mechanical properties prior and after the peak strength, Hou [3] proposed a constitutive model for rock strengthening effect around the roadway with rock bolts. Kang et al. [4] described the mechanisms and technologies of rock bolt supporting system in China. The authors suggested that high strength-stiffness-reliability and low-density proactive rock bolts can improve the strength of surrounding rock. Wang et al. [5] derived the equations of shear strength prior *to* and after anchorage. The analysis showed that once the grouting is completed, the shear strength of anchorage range was improved together with the mechanical properties of surrounding rock (elastic modulus, cohesion, friction angle and dilation angle). Wei et al. [6] carried out physical simulation tests to study the influence of pre-tension on anchor strength and deformation and found that high pre-tension can improve the anchorage strength. Based on the triaxial physical simulation tests,

*Author for correspondence (wangxiangyu_cumt@163.com).

†Present address: Nanhu campus, China University of mining and technology, No.1, University Road, Xuzhou City, Jiangsu Province, China

Wang et al. [7] revealed that the rock bolt does not only improve the capacity of fracture surrounding rock, but also controls the discontinuous deformation of fractured surrounding rock. This will in turn improve the continuity and integrity of surrounding rock mass of deep roadway.

In numerical simulation, the bolt section in anchorage range can be represented by material model as well as structural elements. Both methods have been proven successful in simulating the mechanical properties of bolts and anchors under various loading conditions. Meng [8] imported the constitutive model of anchor in fractured rock to FLAC3D and subsequently simulated the effect of anchoring condition on ultimate bearing capacity. Results showed that the higher pre-tension and density can improve the ultimate bearing capacity. Zuo et al. [9] used elastic-isotropic model in FLAC3D to study the influence of the alignment between bolt and borehole dimeters on anchorage effect. Numerical simulation suggested a reasonable alignment (4 – 11 mm diameter difference) can improve the anchorage effect. ~~D.A.~~ Ho et al. [10] used elasto-plastic model to study the fracture mechanism of interface between rock bolt and anchoring agent under various horizontal stress conditions. Thereby, Grasselli [11], Aziz and Jalalifar [12] and Tatone et al. [13] also used material model to simulate various laboratory condition of rock bolt. However, the mesh size is an important factor which can influence the simulation results using material model. This constrains the preciseness of the simulation. On the other hand, structural element approach can overcome this problem. At present, the structural element can be divided into 'cable element' and 'rockbolt element'. Zhang et al. [14] used cable element to study the parameters including length, anchorage length and rock anchor spacing via FLAC3D and proposed several suggestions to support bolt design. Wu et al. [15] attempted to simulate the supporting element (e.g. wooden cylinder) in supported roadway using Cable element and found that the rock bolts can improve the capacity of wooden cylinder and prevent the fracture development. Pull-test or underground excavation with rock bolt support were simulated by (Vardakos et al. [16], Malmgren and Nordlund [17], Li et al. [18], Gao et al. [19], Shreedharan and Kulatilake [20]). Ma et al. [21] used rockbolt element to simulate pull-test via FLAC2D and studied the relationship between rock bolt and rock as well as the mechanical properties of newly developed rock bolt and its preferable conditions.

According to the previous studies, it can be found that the focus was mainly on the anchorage technique, rock bolt, coordination effect and stress transfer. Although there is an overall agreement that high pre-tension can improve the mechanical properties of anchor, the influence of high pre-tension under various rock strength conditions on anchorage was not investigated. Majority of the research analysed the anchorage effect via elasto-plastic model and stress-strain relationship, while there were limited studies ^{investigating} investigated the problem using energy conservation and the fracture evolution of anchorage is a process of energy dissipation and release. Hence, it is more reasonable to study the anchorage effect using energy law (Xie et al. [22]). Many numerical simulation use cable element to approach this topic; this is inappropriate considering the method is one-dimensional element which consists of two degrees of freedom. On the other hand, rockbolt element is more suitable for the analysis as it is a two-dimensional element with three degrees of freedom (Itasca Consulting Group Inc. [23]). The paper used rockbolt element through UDEC Trigon to study the influence of high pre-tension on anchor with various surrounding rock strengths under uniaxial compression. This aims to study the different strengthening behaviours of tension rock bolt under various conditions based on monitor of the energy evolution, the stress-strain relation as well as the fracture development in different rock strengths.

3. Energy balance and components

Nowadays, the study of energy evolution in rock mechanics has been refined to different depth of cover, loading conditions, in-situ stress magnitudes (Salamon [24], Napier [25], Onur Vardar et al. [26], Zhang et al. [27], Wang et al. [28]). In UDEC, the energy exchanged based on the work done on rock, joint and boundary. Without considering dynamic calculation, the energy balance in UDEC mainly involves work done on boundary, stored strain energy and dissipated energy (Itasca Consulting Group Inc. [23]). This study determined the constitutive law based on the previous studies as well as the simulation conditions.

In uniaxial compression, work done from external force is denoted by W , U^e is used to represent the elastic strain energy due to elastic deformation of the anchor. The difference in work done between boundary and elastic strain energy is dissipated energy, U^d . Hence, total input energy can be written as:

$$W = U^d + U^e \quad (1)$$

Energy is mainly dissipated in three ways. Firstly, energy is dissipated by friction due to the fracture development in rock; this is denoted as W_f . Second source is due to the plastic deformation of rock (W_p). Once the rock is undergone plastic deformation, energy dissipated from plastic work done. The rest of energy is dissipated via acoustic emission, denoted as U_r . Therefore, U^d can be expressed as:

$$U^d = W_f + W_p + U^r \quad (2)$$

In UDEC, the incremental change of these energy components is cumulative and determined by timesteps (Itasca Consulting Group Inc. [23]). By combining Eq (1) and Eq (2), W can be expressed as:

$$W = W_f + W_p + U^r + U^e \quad (3)$$

Subsequently, this energy balance equation can be used to investigate the complex energy dissipation process in anchor.

4. Parameters calibration

4.1. The UDEC Trigon approach

UDEC Trigon model was proposed by Gao and Stead [29], aiming at simulating brittle fracture of rock. In this model, a rock is represented by an assembly of triangular blocks bonded together via the grain contacts. Each block is made elastically by dividing them into triangular finite difference zones. In the direction normal to a contact, the stress-displacement relation is assumed to be linear and governed by the stiffness k_n , such that:

$$\Delta\sigma_n = -k_n \Delta u_n \quad (4)$$

where $\Delta\sigma_n$ is the effective normal stress increment and Δu_n is the normal displacement increment. A limiting tensile strength (T) is assumed for the contact. If this value is exceeded, then $\sigma_n = 0$.

In the shear direction, the response is governed by constant shear stiffness. The shear stress, τ_s , is determined by a combination of contact properties, cohesion (c) and friction angle (ϕ), where:

$$|\tau_s| \leq c + \sigma_n \tan\phi = \tau_{max} \quad (5)$$

then

$$\forall \tau_s = -k_s \forall u_s^e \quad (6)$$

however, if $|\tau_s| \geq \tau_{max}$, then:

$$\tau_s = \text{sign}(\Delta u_s^e) \tau_{max} \quad (7)$$

where Δu_s^e is the elastic component of the incremental shear displacement and Δu_s is the total incremental shear displacement.

The proposed modeling approach has been implemented in UDEC (Itasca Consulting Group Inc.[23]).

4.2 Mechanical parameters of coal and rock mass

Since blocks in UDEC Trigon model are considered as elastic materials, they cannot be plastically destroyed. However, the anchor breakage is plastic damage. To closely mimic the mechanical properties and energy evolution of anchor post peak strength, this study also used strain softening model in UDEC. The strain-softening model is based on the UDEC Mohr–Coulomb model with non-associated shear and associated tension flow rules.

This study selected coal (soft), sandy mudstone (medium) and sandstone (strong) to represent various rock strengths. The intact properties of the rock masses are listed in Table 1. These properties were obtained through laboratory compression tests and were provided by the Yuwu coal mine. The RQD values of the rock masses were evaluated from borehole televiewer images.

Table 1 Intact rock properties and scaled rock mass properties of coal measures from Yuwu coal mine

The rock mass elastic modulus was calculated using the relationship between RQD and the elastic modulus ratio (Zhang and Einstein [30]), as shown in Eq. 8, where E_m is the elastic modulus of the rock mass and E_r is the elastic modulus of the rock sample.

$$\frac{E_m}{E_r} = 10^{0.0186RQD - 1.91} \quad (8)$$

The rock mass strength was then calculated using the relation between the unconfined compressive strength ratio σ_{cm} / σ_c and the deformation modulus ratio E_m / E_r (Singh and Seshagiri Rao [31]). The value of q is 0.63 (Gao. et al, [32]):

$$\frac{\sigma_{cm}}{\sigma_c} = \left(\frac{E_m}{E_r} \right)^q \quad (9)$$

To represent the coal and rock by using an assembly of triangular blocks, the properties of the blocks and contacts were calibrated against the coal measure and rock mass properties listed in Table 2 to 4. This was

achieved by simulating unconfined compression tests in a numerical model created using the Trigon logic. The size of the rock sample is 1 m (in width) × 2 m (in height) (Singh and Seshagiri Rao [31]) (Fig. 1). The bottom of the numerical model was fixed and a loading rate of 0.02 m/s was applied at the top. The calibrated properties of the UDEC model are illustrated in Table 5.

The uniaxial compressive strength and elastic modulus data derived by numerical simulation are similar to the data obtained from laboratory tests (within an error of 9%). Hence, the micromechanical parameters of the coal mass and the rock mass were properly calibrated.

will be good to include this Fig.

Table 2 Calibrated mechanical parameters of blocks and joints of the coal measures

Table 3 Calibrated mechanical parameters of blocks and joints of sandy mudstone

Table 4 Calibrated mechanical parameters of blocks and joints of sandstone

Table 5 Calibrated microproperties in the UDEC Trigon model to represent the rock masses

Fig.1 Results of uniaxial compressive strength tests a numerical simulation of uniaxial compression, b stress–strain curve of coal, c stress–strain curve of sandy mudstone, d stress–strain curve of sandstone

3.3 Rockbolt element parameter

In UDEC, rockbolt element is different from cable element. Rockbolt element is a two-dimensional element which its two nodes have three degrees of freedom (two displacements and one rotation). It can resist bending and can yield along axial direction. Fig. 2 illustrates the components of rockbolt element. Rockbolts interact with UDEC via shear and normal coupling springs, which the springs are non-linear connectors. These are used to transfer force and motion between rockbolt element and the mesh nodes. Fig. 2 shows material behavior of shear and normal coupling spring for rockbolt elements. Shear coupling springs are used to simulate the shear behaviour of anchorage length and normal coupling spring are used to simulate the compression of surrounding rock.

Fig.2 Conceptual mechanical representation of the rockbolt element

Rockbolt element can be broken at the node. The bolt breakage can be simulated by defining the tensile failure strain limit ($tfstrain$). Rock bolt will be deemed as failed if $\varepsilon_{pl} \geq tfstrain$, where $tfstrain$ has to be defined. ε_{pl} is the total tensile yield strain of any element of anchor:

$$\varepsilon_{pl} = \sum \varepsilon_{pl}^{ax} + \sum \frac{d}{2} \frac{\theta_{pl}}{L} \quad (10)$$

ε_{pl}^{ax} is the axial deformation; d is rock bolt diameter, θ_{pl} is the average rotation angle of the component.

This study used rockbolt element to simulate HRB335 threaded steel anchor at 22 mm diameter. Based on the laboratory pull-test results, the input parameters of rockbolt element can be obtained, as displayed in Table 6.

Table 6 Rockbolt and anchor parameters

5. Research background and model establishment

The common way to apply pre-tension is to install the nut on the tail thread of rock bolt. By applying torque to the nut, axial tension is acting onto the rock bolt (Li et al. [33]). However, high-pretension is difficult to be controlled in practice. For instance, when the torque achieves 300 – 400 Nm, the pre-tension can sometimes still be low. This low conversion efficiency is due to the torque at thread. To increase the pre-tension of rock bolt, a new type of anchor lock has been developed by the research team. This development changed the traditional pre-tension method, which is capable of applying high pre-tension without damaging rock bolt. The anchor lock is composed of an anchor ring and three clips. The anchor ring is a ring-shaped structure, in which the center hole is tapered. The clips are disposed between the inner wall of the tapered hole of the anchor ring and the anchor body, as displayed in Fig. 3.

why diff. spacing?

Part of this paper. Ref?

Fig.3 Rockbolt barrel and wedge and assembly drawing

Fig. 4 shows the setup of UDEC Trigon simulation. The blocks and interfaces used strain softening and Coulomb-sliding models, respectively. The dimensions of the numerical model are 1 m × 2m. Due to the

limitation of rockbolt element, the effect of vertically installed rock bolts was not significant. Hence, rock bolts were installed horizontally to the rock body at $y = 0.4\text{m}$, 1.0m and 1.6m . Pre-tension was applied by applying load at the nodes of the rock bolts. In practice, the common torque applied to the rock bolt is approximately $300 - 400\text{ Nm}$, which are $40 - 50\text{ kN}$ pre-tension correspondingly (Kang et al. [34]). To achieve high pre-tension in the numerical model, the study set the pre-tension as 70% of the yielding strength, i.e. $138.6\text{ kN}/0.365\text{ MPa}$. The loading rate of 0.02 m/s was applied at the top boundary of the model while fixing the bottom boundary. During the simulation process, the stress-strain relationship and fracture development of anchor was monitored by FISH function. Boundary, frictional and plastic work were recorded using UDEC energy element. 'SET energy on' command was used to activate the energy monitoring function while turn off 'mass-scaling'. As this study used non-viscous boundary which does not consider the dynamic calculation, damping was set as 'auto'. The simulation procedures involved: i) model construction and calibration; ii) test simulation.

Table 7 Numerical simulation scheme

Fig.4 Model overview and dimensions

6. Discussion

6.1 Mechanical properties and damage analysis

To monitor the degree of damage of rock and anchor, the total fracture length as well as the shear and tensile fracture lengths during uniaxial compression were measured. The anchor and surrounding rock fracture are a complex process, which the shear and tensile failures can occur in a single fracture. Hence, only the initial (first) failure type was recorded for each fracture. The degree of damage (D) can be calculated as:

$$D = \frac{L_s + L_t}{L_c} \times 100\% \quad (11)$$

where L_c is the total fracture length, L_s and L_t are total shear and tensile fracture lengths, respectively.

Fig.5 The stress and degree of damage of rock with and without anchorage a coal without anchorage; b coal with anchorage; c sandy mudstone without anchorage; d sandy mudstone with anchorage; e sandstone without anchorage; f sandstone with anchorage

Fig. 5 illustrates the stress and degree of damage of rock during the process with and without anchorage. Based on the comparison, the mechanical properties (peak strength, residual strength and elastic modulus) of rock were increased with the installation of tension rock bolts, while the degree of damage decreased. Under the same axial strain, the higher strength always led to higher degree of damage regardless of the anchorage condition. The installation of tension rock bolts did not have significant influence on the axial strain of rock during damage. There was increase in the degree of damage prior to the peak rock strength and the gradient became flatter when it was near the peak strength. The indication here is that installation of tension bolts can prevent the fracture development and coalescence, such that the degree of damage was reduced. To compare the effect of tension rock bolts under different rock conditions, Table 8 summaries the critical parameters from Fig. 5.

Table 8 Mechanical properties and degree of damage of rock prior and after anchorage

Table 8 suggested that there are differences in the strengthening effect of tension rock bolts on rock mechanical properties with various rock types. When the bolt was installed in coal, elastic modulus, peak strength and residual strength increased 4.1% (0.03 GPa), 5.9% (0.3 MPa) and 5.2% (0.2 MPa), while the degree of damage decreased 13.7%. In sandy mudstone, elastic modulus, peak strength and residual strength increased 20.7% (0.6 GPa), 3.6% (0.9 MPa) and 13.1% (1.7MPa), while the degree of damage decreased 11.9%. For sandstone, elastic modulus, peak strength and residual strength increased 21.1% (0.1 GPa), 0.6% (0.2 MPa) and 0.5% (0.1 MPa), while the degree of damage decreased 5.3%. Sandy mudstone had the most change in absolute magnitudes and sandstone had the least change. In terms of the relative magnitude difference, coal had the most change whereas sandstone had the least. The absolute magnitude change of sandy mudstone was the most while its degree of damage was only in the middle, indicating the mechanical properties of rock was improved by installation of rock bolts. However, this is not the primary reason for the reduction in degree of damage, in which the main reason was due to the prevention of fracture development and coalescence using rock bolts. Sandstone had the low change in terms of absolute and relative magnitudes. This suggests that high pre-tension rock bolts do not have significant influence on strong rock. This is because the rock strength was high and it required substantially higher stress to induce the rock failure. Under this condition, the effect of rock bolt is negligible comparing with the high strength. At post-peak of the loading process, the

fracture was more likely to be developed due to the low plasticity, which was difficult to be controlled using rock bolts. Overall, it can be concluded that tension rock bolts have insignificant influence on strong rock, such that the high pre-tension is not recommended for hard rock support design.

6.2 Analysis of the law of energy evolution

During the loading process, rock was continuously deforming and storing elastic strain energy. At the same time, the energy was dissipated due to internal damage. When the elastic strain energy reached the limit, it was released and led to rock failure. Hence, the failure process of rock is closely related to the energy evolution. Based on the analysis carried out in Section 5.1, tension rock bolts are not suitable for hard rock. Therefore, this section will focus on the energy evolution of coal and sandy mudstone, as displayed in Fig. 6 and 7.

Fig.6 Stress and energy evolution with and without anchorage in coal a stress-strain curve of coal (without anchorage); b a stress-strain curve of coal (without anchorage); c energy evolution of coal (without anchorage); d energy evolution of coal (anchorage)

Fig.7 Stress and energy evolution with and without anchorage in sandy mudstone a stress-strain curve of sandy mudstone (without anchorage); b a stress-strain curve of sandy mudstone (without anchorage); c energy evolution of sandy mudstone (without anchorage); d energy evolution of sandy mudstone (anchorage)

In Fig. 6 and 7, the energy dissipation due to acoustic emission (U^r) was illustrated solely. This is because the energy dissipation curve of acoustic emission was similar as the accumulative acoustic emission, which is a major indication of anchor breakage. Secondly, the energy dissipation due to acoustic emission was substantially lower than the other sources. A sole illustration can represent the energy evolution of acoustic emission more clearly. According to the results from Fig. 6 and 7, the energy evolution of rock was consistent regardless of the rock bolts. Based on the correspondence between stress curve and energy curve, the energy evolution of rock can be divided into three stages:

Stage 1: From the start of the load to the yielding strength. External work was fully absorbed by rock and stored in the form of elastic strain energy. The energy curve increased non-linearly during this stage, while the external work curve was well aligned with the elastic strain curve. Based on the stress and energy curves, the rock was under elastic deformation prior to the yielding strength. There was no observable internal damage, such that the internal energy dissipation was considered as none.

Stage 2: From the yielding strength to the peak strength. External work continuously increased and the total energy that the rock absorbed was also increasing. However, as the rock was in plastic condition and there was some degree of internal damage, the frictional work in-between cracks and plastic deformation started to occur and increased at a high rate. This indicates that the degree of damage was decreased although the acoustic emission level was still low. During this stage, the elastic strain energy was also increasing but the growth rate was flatter. Thereby, its alignment with the external work was getting worse and reached its maximum near the peak strength point.

Stage 3: Post-peak stage. External work continuously increased but the elastic energy in the rock was released at a high rate. There was considerable increase in the energy dissipation due to frictional work, plastic deformation and acoustic emission. The acoustic mission energy curve was growing rapidly, suggesting that the degree of damage was high. The external work was mainly transferred to the energy require for rock fracture towards the end of the simulation and the elastic strain energy was not stored anymore. When the stress dropped to the residual strength, the elastic strain energy and energy dissipation curves became flatter, although the energy continuously dissipated.

Although the energy evolution of rock was consistent regardless of the existence of rock bolts, the degree of damage and other parameters changed noticeably with the installation of rock bolts. Firstly, there were increasing in stress and decreasing in strain at anchor damage with rock bolts. For coal, the stress increased from 4.4 MPa to 4.8 MPa and strain reduced from 0.57% to 0.5%. For sandy mudstone, stress increased from 18 MPa to 20.6 MPa and strain reduced from 0.6% to 0.49%. Secondly, the peak elastic energy increased from 29.2 kJ to 35 kJ and from 202.7 kJ to 240.7 kJ for coal and sandy mudstone, respectively. When strain reached 1.2%, the external work of coal and sandy mudstone increased 26.2 kJ and 33.7 kJ. For coal, the work done from fracture friction was 35.1 kJ without rock bolt, accounting for 50.1% of the total input energy; whereas the energy was 37.4 kJ and took account of 38.9% with rock bolt. On the other hand, for sandy mudstone, the work done from fracture friction was 230.3 kJ without rock bolt, accounting for 58.8% of the total input energy; whereas the energy was 158.7 kJ and took account of 37.2% with rock bolt. Based on aforementioned data, the installation of tension rock bolts can improve the strength of surrounding rock and its storage of elastic strain energy. At the same time, it can also prevent fracture development and postpone the elastic strain energy release at post-peak. Fig. 8 shows the comparison of each energy categories. From the figure, it can be seen that the performance of sandy mudstone was better than coal. Hence, the indication here is that the tension

Isn't this a strong statement to make based on this limited study. I would suggest to rephrase.

rock bolts can improve the mechanical properties and energy storage capacity of rock, which agrees with the conclusion from Section 6.1. Thereby, the most suitable rock type for the application is medium hard rock, e.g. sandy mudstone.

Fig.8 The comparison between coal and sandy mudstone on different energy categories

Fig.9 The stress and energy evolution of sandstone with and without anchorage **a** stress-strain relationship curve of sandstone (without anchorage); **b** stress-strain relationship curve of sandstone (with anchorage); **c** energy evolution of sandy mudstone (without anchorage); **d** energy evolution of sandy mudstone (with anchorage)

Fig. 9 depicts the stress and energy evolution of sandstone with and without anchorage. According to the figure, it can be found that the relationships of sandstone follow the same trend as those of coal. The difference between sandstone and coal was that the changes of frictional work and plastic deformation of sandstone were insignificant with rock bolts, indicating that the installation of rock bolts does not increase the plasticity of sandstone nor prevent fracture development. The plastic deformation curve was always at low level while the frictional work curve was always at high level. This phenomenon again supports that the fracture development was the main reason of rock failure. This agrees with the conclusion from stress and damage analysis; thereby proved that tension rock bolts are not effective for hard rock roadway support, e.g. sandstone.

7 Conclusion

Based on the UDEC simulation on the strengthening effect of tension rock bolts in different rock properties, a number of parameters were studied, including mechanical properties, degree of damage and energy evolution. The following conclusions were drawn from the numerical results:

(1) Tension rock bolts can improve the mechanical properties and energy storage capacity of anchor. Subsequently, it can reduce the degree of damage of the anchor while increasing the elastic strain energy storage and plastic deformation work. By suppressing energy dissipation due to fracture friction and delaying the post-peak elastic strain release, the stability of anchor can be increased.

(2) Based on the stress-strain and energy dissipation-strain relationships during uniaxial compression, the energy evolution of anchor was divided into three stages. In the first stage, the external work was fully absorbed by rock and was stored as elastic deformation energy. In the second stage, the axial stress surpasses the rock yielding strength, which work was done from fracture friction and plastic deformation. At the same time, the growth rate of elastic strain energy was reduced and the elastic strain energy reached the maximum value near the peak strength point. In the third stage, the energy stored in the rock was rapidly released from elastic strain energy. The work from fracture friction, plastic deformation and acoustic emission increased substantially. At the end of the process, the energy of external work was mainly transferred to rock failure, where the energy was dissipated via fracture friction, plastic deformation and acoustic emission. Among them, work from fracture friction and plastic deformation was the major contributing factor for energy dissipation.

(3) Based on stress and energy evolution analysis on different rock types, it can be seen that the tension rock bolts were not effective for hard rock. The elastic modulus, peak strength and residual strength increased 2.1% (0.1 GPa), 0.6% (0.2 MPa) and 0.5% (0.1 MPa), respectively. The implementation could not prevent the fracture development within the rock mass, such that it was not recommended for hard rock supporting design.

(4) The absolute magnitudes of mechanical properties of sandy mudstone increased more than that of coal. However, the relative magnitudes and reduction of degree of damage was lower than coal. This indicates that the improvement of mechanical properties of rock due to rock bolts was the secondary factor for degree of damage reduction, whereas its effect on preventing fracture development and coalescence was the main contributing factor. For energy evolution, the stress at initial damage increased 14.4% (2.6 MPa) and 9% (0.4 MPa) for sandy mudstone and coal. In addition, the peak elastic strain energy increased 18.7% (38 kJ) and 9.6% (5.8 kJ) for sandy mudstone and coal. The work done due to fracture friction decreased from 58.8% to 37.2% of the total energy for sandy mudstone; and it decreased from 50.1% to 38.9% for coal. Therefore, tension rock bolts is more suitable for medium hard rock (e.g. sandy mudstone), followed by soft rock (e.g. coal).

Acknowledgments

The authors are grateful for the valuable comments from the editors and the reviewer, which has substantially improved the quality of our work.

Funding Statement

This work is supported by the Fundamental Research Funds for the Central Universities (2019BSCX05).

Data Accessibility

Our data are deposited at Dryad.

Review URL: <https://datadryad.org/stash/share/Agf-g6O9idvPjujUn9zqr8m79qBwHC6MF69F5n4fWk8>

doi:10.5061/dryad.cvdncjt1w

Competing Interests

We have no competing interests.

Authors' Contributions

B.W. and X.W. conceived the article structure. B.W. and W.W. performed the rock mechanical experiments. J.B. developed energy balance criterion. B.W. numerical simulation. All the authors analysed the data. N.M. prepared the plots. H.L. language revision. All the authors prepared the initial draft and revised the paper.

References

REF SHOULD BE CONSISTENT.

- 1 Navid Bahrani, John Hadji Georgiou 2017 (https://doi.org/10.1016/j.ijrmmms.2004.06.003) 1922. (https://doi.org/10.1007/s00603-015-0885-9)
- 2 Wen ZJ, Xing ER, Shi SS et al. 2018 Overlying strata structural modeling and support applicability analysis for large mining-height stopes. *J. Loss Prevent Proc.* **57**,94-100. (https://doi.org/10.1016/j.jlpp.2018.11.006)
- 3 Hou CJ, Gou PF 2000 Study on the strengthening mechanism of surrounding rock strength with bolting support. *Chin J Rock Mec Eng.* **19**, 342-345.
- 4 Kang HP, Jiang TM, Gao FQ. Effect of pretension stress to rock bolting. *J Coal Soc.* **2007**;32(7):680-685.
- 5 Wang Q, Qin Q, Jiang B et al. 2019 Study and engineering application on the bolt-grouting reinforcement effect in underground engineering with fractured surrounding rock. *Tunn. Undergr. Space Technol.* **84**,237-247. (https://doi.org/10.1016/j.tust.2018.11.028)
- 6 Wei SJ, Gou PF 2012 Analogy simulation test on strengthening effect for pretention of bolts on anchorage body. *J China Coal Soc.* **37**, 1987-1993.
- 7 Wang WJ, Dong EY, Zhao ZW et al. 2020 Experimental study on mechanical properties of anchorage body and on anchorage mechanism. *J China Coal Soc.* **45**,82-89.
- 8 Meng B. 2013 Study on bearing characteristic of soft and fractured roadway surrounding rock anchorage unit and its applications in engineering. Ph.D. thesis, China University of Mining and Technology.
- 9 Zuo JP, Wen JH, Li YD et al. 2019 Investigation on the interaction mechanism and failure behavior between bolt and rock-like mass. *Tunn. Undergr. Space Technol.* **93**:14. (https://doi.org/10.1016/j.tust.2019.103070)
- 10 D.-A. Ho, M. Bost, J.-P. Rajo 2019 Numerical study of the bolt-grout interface for fully grouted rockbolt under different confining conditions. *Int J Rock Mech Min Sci.* **119**,168-179. (https://doi.org/10.1016/j.ijrmmms.2019.04.017)
- 11 Grasselli G. 2005 3D behaviour of bolted rock joints: experimental and numerical study. *Intel J Rock Mech Min Sci.* **42**,13-24.
- 12 Aziz N, Jalalifar H. 2007 Experimental and numerical study of double shearing of bolt under confinement. In: Peng SS, Mark C, Finfinger G, Tadolini S, Khair AW, Heasley K, Luo Y, editors. Proceedings of the 26th International Conference on Ground Control in Mining, Morgantown, WV, USA.
- 13 Tatone BSA, Lisjak A, Mahabadi OK, et al. 2015 Incorporating rock reinforcement elements into numerical analysis based on the hybrid finite-discrete element method (DFEM). In: Proceedings of ISRM Congress, Montreal, Canada.
- 14 Zhang K, Zhang G, Hou R et al. 2015 Stress evolution in roadway rock bolts during mining in a fully mechanized longwall face and an evaluation of rock bolt support design. *Rock Mech. Rock Eng.* **48**, 333-344.
- 15 Wu BW, Wang XY, Bai JB et al. 2019 Study on crack evolution mechanism of roadside backfill body in gob-side entry retaining based on UDEC Trigon model. *Rock Mech Rock Eng.* **52**,3385-3399. (https://doi.org/10.1007/s00603-019-01789-6)
- 16 Vardakos SS, Gutierrez MS, Barton NR. 2007 Back-analysis of Shimizu Tunnel No. 3 by distinct element modeling. *Tunn. Undergr. Space Technol.* **22**,401-413. (https://doi.org/10.1016/j.tust.2006.10.01)
- 17 Malmgren L, Nordlund E. 2008 Interaction of shotcrete with rock and rock bolts: a numerical study. *Inter J Rock Mech Min Sci.* **45**,538-553. (https://doi.org/10.1016/j.ijrmmms.2007.07.024)
- 18 Li B, Qi T, Wang ZZ, Yang L 2012 Back analysis of grouted rock bolt pullout strength parameters from field tests. *Tunn Undergr. Space Technol.* **28**,345-349. (https://doi.org/10.1016/j.tust.2011.11.004)
- 19 Gao F, Stead D, Kang H. 2015 Numerical simulation of squeezing failure in a coal mine roadway due to mining-induced stresses. *Rock Mech Rock Eng.* **48**,1635-1645. (https://doi.org/10.1007/s00603-014-0653-2)
- 20 Shreedharan S, Kulatilake PHSW 2016 Discontinuum-equivalent continuum analysis of the stability of tunnels in a deep coal mine using the distinct element method. *Rock Mech Rock Eng.* **49**,1903-1922. (https://doi.org/10.1007/s00603-015-0885-9)
- 21 Ma S, Nemcik J, Aziz N 2014 Simulation of fully grouted rockbolts in underground roadways using FLAC2D. *Canadian Geotech J.* **51**,911-920. (https://doi.org/10.1139/cgj-2018-0470)
- 22 Xie HP, Ju Y, Li LY et al. 2008 Energy mechanism of deformation and failure of rock masses based on energy dissipation and energy release principles. *Chin J Rock Mech Eng.* **27**,1729-1938c.
- 23 Itasca Consulting Group Inc.2014 UDEC (Universal Distinct Element Code), Version 6.0. Itasca, Minneapolis.
- 24 Salamon MDG. 1984 Energy considerations in rock mechanics: fundamental results. *J S Afr Inst Min Metall.* **84**,233-46.
- 25 Napier JAL.1991 Energy changes in a rockmass containing multiple discontinuities. *J S Afr Inst Min Metall.* **91**,145-57.
- 26 Onur Vardar, Zhang CG, Ismet Canbulat et al.2019 Numerical modelling of strength and energy release characteristics of pillar-scale coal mass. *J Rock Mec Geotech Eng.* **11**,935-943. (https://doi.org/10.1016/j.jrmge.2019.04.003)
- 27 Zhang ZZ, Deng M, Bai JB et al. 2020 Strain energy evolution and conversion under triaxial unloading confining pressure tests due to gob-side entry retained. *Inteal J Rock Mech Min Sci.* **126**:104184. (doi:10.1016/j.ijrmmms.2019.104184)
- 28 Wang M, Zheng D, Wang K, Li W. 2018 Strain energy analysis of floor heave in longwall gateroads. *R. Soc. open sci.* **5**: 180691. (http://dx.doi.org/10.1098/rsos.180691)
- 29 Gao FQ, Stead D, Coggan J. 2014 Evaluation of coal longwall caving characteristics using an innovative udec trigon approach. *Comput Geotech.* **55**,448-460. (https://doi.org/10.1016/j.compgeo.2013.09.020)
- 30 Zhang L, Einstein HH 2004 Using RQD to estimate the deformation modulus of rock masses. *Int J Rock Mech Min Sci.* **41**,337-341. (https://doi.org/10.1016/S1365-1609(03)00100-X)
- 31 Singh M, Rao KS. 2005 Empirical methods to estimate the strength of jointed rock masses. *Eng Geol.* **77**,127-137. (https://doi.org/10.1016/j.enggeo.2004.09.001)
- 32 Gao FQ, Stead D. 2014 The application of a modified voronoi logic to brittle fracture modelling at the laboratory and field

scale. *Int J Rock Mech Min Sci.* **68**,1–14. (https://doi.org/10.1016/j.ijrmms.2014.02.003)

33 Li ZB, Zhang N, Han Cl et al. 2012 Relationship between pre-tightening force and tightening torque. *J China U Min Technol.* **41**,189-193.

34 Kang HP, Wang Y, Gao FQ. 2016 Mechanical performances and stress states of rock bolts under varying loading conditions. *Tunn. Undergr. Space Technol.* **52**,138-146. (https://doi.org/10.1016/j.tust.2015.12.005)

35 Wang, Xiangyu (2020), An Investigation on the effect of high energy storage anchor on surrounding rock conditions, v3, Dryad, Dataset, <https://doi.org/10.5061/dryad.cvdncjt1w>

Tables

Table 1 Intact rock properties and scaled rock mass properties of coal measures from Yuwu coal mine

Lithology	Intact rock		RQD	Rock mass		
	E_r (GPa)	$\bar{\sigma}_r$ (MPa)		E_m (GPa)	$\bar{\sigma}_{cm}$ (MPa)	$\bar{\sigma}_{tm}$ (MPa)
Coal	2.6	10.8	75%	0.79	5.1	0.51
Sandy mudstone	5.4	35.5	90%	3.1	25	2.5
Sandstone	9.7	53.5	88%	5.2	36.1	3.6

Table 2 Calibrated mechanical parameters of blocks and joints of the coal measures

	Density (kg/m ³)	Young's modulus (GPa)	Cohesion (MPa)	Friction (°)	Tension (MPa)
Block	1400	0.79	1.6 ($\epsilon_p=0$) 1.1 ($\epsilon_p=0.04$) 0.6 ($\epsilon_p=0.15$)	27	0.9
Joint	Normal stiffness (GPa)	Tangential stiffness (GPa)	Cohesion (MPa)	Friction (°)	Tension (MPa)
	113	45.2	1.3	18	0.4

Table 3 Calibrated mechanical parameters of blocks and joints of sandy mudstone

	Density (kg/m ³)	Young's modulus (GPa)	Cohesion (MPa)	Friction (°)	Tension (MPa)
Block	1800	3.1	8.0 ($\epsilon_p=0$) 5.0 ($\epsilon_p=0.06$) 2.0 ($\epsilon_p=0.15$)	27	2.5
Joint	Normal stiffness (GPa)	Tangential stiffness (GPa)	Cohesion (MPa)	Friction (°)	Tension (MPa)
	372	149	6.2	18	1.8

Table 4 Calibrated mechanical parameters of blocks and joints of sandstone

	Density (kg/m ³)	Young's modulus (GPa)	Cohesion (MPa)	Friction (°)	Tension (MPa)
Block	2550	5.2	13.0 ($\epsilon_p=0$) 7.0 ($\epsilon_p=0.04$) 4.0 ($\epsilon_p=0.10$)	30	2.5
Joint	Normal stiffness (GPa)	Tangential stiffness (GPa)	Cohesion (MPa)	Friction (°)	Tension (MPa)
	665	266	11.2	18	1.8

Table 5 Calibrated microproperties in the UDEC Trigon model to represent the rock masses

Lithology	Young's modulus(GPa)			Compressive strength(MPa)		
	Target	Calibrated	Error(%)	Target	Calibrated	Error(%)
Coal	0.79	0.73	7	5.1	5.1	0
Sandy mudstone	3.1	2.9	6	25	25.1	0.4
Sandstone	5.2	4.7	9	36.1	36.3	5

Table 6 Rockbolt and anchor parameters

	Cross-section area (m ²)	Elastic modulus (GPa)	Tensile yield strength (kN)	Second moment of area(m ⁴)	tfstrain		
Rockbolt	3.8×10^{-4}	200	198	1.2×10^{-8}	0.012		
	Exposed perimeter (m)	Cohesive strength of shear coupling spring (MPa)	Stiffness of shear coupling spring (GPa)	Frictional resistance of the shear coupling spring (°)	Cohesive strength of normal coupling spring (MPa)	Stiffness of normal coupling spring (GPa)	Frictional resistance of the normal coupling spring (°)
Anchor parameters	0.07	1	8	45	200	20	0

Table 7 Numerical simulation scheme

	Rockbolt existence	Pre-tension (load)	Loading rate
Coal (soft)	No	/	0.02m/s
	Yes	0.365MPa	0.02m/s
Sandy mudstone (medium)	No	/	0.02m/s
	Yes	0.365MPa	0.02m/s
Sandstone (strong)	No	/	0.02m/s
	Yes	0.365MPa	0.02m/s

Table 8 Mechanical properties and degree of damage of rock prior and after anchorage

	Rock bolt existence	Elastic modulus (GPa)	Peak strength (MPa)	Residual strength (MPa)	Degree of damage (%)
Coal (soft)	No	0.73	5.1	3.8	35.1
	Yes	0.76	5.4	4.0	30.3
Sandy mudstone (medium)	No	2.9	25.1	13	46.2
	Yes	3.5	26	14.7	40.7
Sandstone (strong)	No	4.7	36.3	20.2	50.8
	Yes	4.8	36.5	20.3	48.1

Figure and table captions

Table.1 Intact rock properties and scaled rock mass properties of coal measures from Yuwu coal mine

Table.2 Calibrated mechanical parameters of blocks and joints of the coal measures

Table.3 Calibrated mechanical parameters of blocks and joints of sandy mudstone

Table.4 Calibrated mechanical parameters of blocks and joints of sandstone

Table.5 Calibrated microproperties in the UDEC Trigon model to represent the rock masses

Table.6 Rockbolt and anchor parameters

Table.7 Numerical simulation scheme

Table.8 Mechanical properties and degree of damage of rock prior and after anchorage

Figure.1 Results of uniaxial compressive strength tests **a** numerical simulation of uniaxial compression, **b** stress - strain curve of coal, **c** stress - strain curve of sandy mudstone, **d** stress - strain curve of sandstone

Figure.2 Conceptual mechanical representation of the rockbolt element

Figure.3 Rockbolt barrel and wedge and assembly drawing

Figure.4 Model overview and dimensions

Figure.5 The stress and degree of damage of rock with and without anchorage **a** coal without anchorage; **b** coal with anchorage; **c** sandy mudstone without anchorage; **d** sandy mudstone with anchorage; **e** sandstone without anchorage; **f** sandstone with anchorage

Figure.6 Stress and energy evolution with and without anchorage in coal **a** stress-strain curve of coal (without anchorage); **b** a stress-strain curve of coal (without anchorage); **c** energy evolution of coal (without anchorage); **d** energy evolution of coal (anchorage)

Figure.7 Stress and energy evolution with and without anchorage in sandy mudstone **a** stress-strain curve of sandy mudstone (without anchorage); **b** a stress-strain curve of sandy mudstone (without anchorage); **c** energy evolution of sandy mudstone (without anchorage); **d** energy evolution of sandy mudstone (anchorage)

Figure.8 The comparison between coal and sandy mudstone on different energy categories

1
2
3 Figure.9 The stress and energy evolution of sandstone with and without anchorage **a** stress-strain relationship
4 curve of sandstone (without anchorage); **b** stress-strain relationship curve of sandstone (with anchorage);
5 **c** energy evolution of sandy mudstone (without anchorage); **d** energy evolution of sandy mudstone (with
6 anchorage)
7
8
9
10
11
12
13
14
15
16
17
18
19
20
21
22
23
24
25
26
27
28
29
30
31
32
33
34
35
36
37
38
39
40
41
42
43
44
45
46
47
48
49
50
51
52
53
54
55
56
57
58
59
60

Fig.1 Results of uniaxial compressive strength tests **a** numerical simulation of uniaxial compression, **b** stress–strain curve of coal, **c** stress–strain curve of sandy mudstone, **d** stress–strain curve of sandstone

Fig.2 Conceptual mechanical representation of the rockbolt element

Fig.3 Rockbolt barrel and wedge and assembly drawing

Was it part of this study? Confused.

Fig.4 Model overview and dimensions

Fig.5 The stress and degree of damage of rock with and without anchorage **a** coal without anchorage; **b** coal with anchorage; **c** sandy mudstone without anchorage; **d** sandy mudstone with anchorage; **e** sandstone without anchorage; **f** sandstone with anchorage

Diff from the fig.

Fig.6 Stress and energy evolution with and without anchorage in coal **a** stress-strain curve of coal (without anchorage); **b** a stress-strain curve of coal (with anchorage); **c** energy evolution of coal (without anchorage); **d** energy evolution of coal (with anchorage)

Fig.7 Stress and energy evolution with and without anchorage in sandy mudstone **a** stress-strain curve of sandy mudstone (without anchorage); **b** a stress-strain curve of sandy mudstone (with anchorage); **c** energy evolution of sandy mudstone (without anchorage); **d** energy evolution of sandy mudstone (with anchorage)

Fig.8 The comparison between coal and sandy mudstone on different energy categories

Fig.9 The stress and energy evolution of sandstone with and without anchorage **a** stress-strain relationship curve of sandstone (without anchorage); **b** stress-strain relationship curve of sandstone (with anchorage); **c** energy evolution of sandy mudstone (without anchorage); **d** energy evolution of sandy mudstone (with anchorage)

Table 1 Intact rock properties and scaled rock mass properties of coal measures from Yuwu coal mine

Lithology	Intact rock		RQD	Rock mass		
	E_r (GPa)	$\bar{\sigma}_r$ (MPa)		E_m (GPa)	$\bar{\sigma}_{cm}$ (MPa)	$\bar{\sigma}_m$ (MPa)
Coal	2.6	10.8	75%	0.79	5.1	0.51
Sandy mudstone	5.4	35.5	90%	3.1	25	2.5
Sandstone	9.7	53.5	88%	5.2	36.1	3.6

Table 2 Calibrated mechanical parameters of blocks and joints of the coal measures

	Density	Young's	Cohesion	Friction	Tension
	(kg/m ³)	modulus (GPa)	(MPa)	(°)	(MPa)
Block	1400	0.79	1.6 ($\epsilon_p=0$)	27	0.9
			1.1 ($\epsilon_p=0.04$)		
			0.6 ($\epsilon_p=0.15$)		
Joint	Normal	Tangential	Cohesion	Friction	Tension
	stiffness (GPa)	stiffness (GPa)	(MPa)	(°)	(MPa)
	113	45.2	1.3	18	0.4

Table 3 Calibrated mechanical parameters of blocks and joints of sandy mudstone

	Density	Young's	Cohesion	Friction	Tension
	(kg/m ³)	modulus (GPa)	(MPa)	(°)	(MPa)
Block	1800	3.1	8.0 ($\epsilon_p=0$)	27	2.5
			5.0 ($\epsilon_p=0.06$)		
			2.0 ($\epsilon_p=0.15$)		
Joint	Normal	Tangential	Cohesion	Friction	Tension
	stiffness (GPa)	stiffness (GPa)	(MPa)	(°)	(MPa)
	372	149	6.2	18	1.8

Table 4 Calibrated mechanical parameters of blocks and joints of sandstone

	Density	Young's	Cohesion	Friction	Tension
	(kg/m ³)	modulus (GPa)	(MPa)	(°)	(MPa)
Block	2550	5.2	13.0 ($\epsilon_p=0$)	30	2.5
			7.0 ($\epsilon_p=0.04$)		
			4.0 ($\epsilon_p=0.10$)		
Joint	Normal	Tangential	Cohesion	Friction	Tension
	stiffness (GPa)	stiffness (GPa)	(MPa)	(°)	(MPa)
	665	266	11.2	18	1.8

Table 5 Calibrated microproperties in the UDEC Trigon model to represent the rock masses

Lithology	Young's modulus(GPa)			Compressive strength(MPa)		
	Target	Calibrated	Error(%)	Target	Calibrated	Error(%)
Coal	0.79	0.73	7	5.1	5.1	0
Sandy mudstone	3.1	2.9	6	25	25.1	0.4
Sandstone	5.2	4.7	9	36.1	36.3	5

Table 6 Rockbolt and anchor parameters

	Cross-section area (m ²)	Elastic modulus (GPa)	Tensile yield strength (kN)	Second moment of area(m ⁴)	tfstrain		
Rockbolt	3.8×10 ⁻⁴	200	198	1.2×10 ⁻⁸	0.012		
	Exposed perimeter (m)	Cohesive strength of shear coupling spring (MPa)	Stiffness of shear coupling spring (GPa)	Frictional resistance of the shear coupling spring (°)	Cohesive strength of normal coupling spring (MPa)	Stiffness of normal coupling spring (GPa)	Frictional resistance of the normal coupling spring (°)
Anchor parameters	0.07	1	8	45	200	20	0

Table 7 Numerical simulation scheme

	Rockbolt existence	Pre-tension (load)	Loading rate
Coal (soft)	No	/	0.02m/s
	Yes	0.365MPa	0.02m/s
Sandy mudstone (medium)	No	/	0.02m/s
	Yes	0.365MPa	0.02m/s
Sandstone (strong)	No	/	0.02m/s
	Yes	0.365MPa	0.02m/s

Table 8 Mechanical properties and degree of damage of rock prior and after anchorage

	Rock bolt existence	Elastic modulus (GPa)	Peak strength (MPa)	Residual strength (MPa)	Degree of damage (%)
Coal (soft)	No	0.73	5.1	3.8	35.1
	Yes	0.76	5.4	4.0	30.3
Sandy mudstone (medium)	No	2.9	25.1	13	46.2
	Yes	3.5	26	14.7	40.7
Sandstone (strong)	No	4.7	36.3	20.2	50.8
	Yes	4.8	36.5	20.3	48.1

Appendix B**ROYAL SOCIETY
OPEN SCIENCE****An Investigation on the effect of high energy storage
anchor on surrounding rock conditions**

Journal:	Royal Society Open Science
Manuscript ID	RSOS-201105
Article Type:	Research
Date Submitted by the Author:	26-Jun-2020
Complete List of Authors:	Wu, Bowen; China University of Mining and Technology; State Key Laboratory of Coal Resources and Safe Mining Wang, Xiang-yu; China University of Mining and Technology; State Key Laboratory of Coal Resources and Safe Mining Bai, Jianbiao; State Key Laboratory of Coal Resources and Safe Mining Wu, Wenda; China University of Mining and Technology Meng, Ningkang; China University of Mining and Technology, School of Mines Lin, Huasheng; University of New South Wales
Subject:	Energy < ENGINEERING AND TECHNOLOGY, Civil engineering < ENGINEERING AND TECHNOLOGY
Keywords:	Energy evolution, tension bolts, UDEC, differential analysis, rock
Subject Category:	Engineering

Author-supplied statements

Relevant information will appear here if provided.

Ethics

Does your article include research that required ethical approval or permits?:

This article does not present research with ethical considerations

Statement (if applicable):

CUST_IF_YES_ETHICS :No data available.

Data

It is a condition of publication that data, code and materials supporting your paper are made publicly available. Does your paper present new data?:

Yes

Statement (if applicable):

Our data are deposited at Dryad.

Review URL: <https://datadryad.org/stash/share/Agf-g6O9idvPjujUn9zqr8m79qBwHC6MF69F5n4fWk8>

doi:10.5061/dryad.cvdncjt1w

Conflict of interest

I/We declare we have no competing interests

Statement (if applicable):

We have no competing interests.

Authors' contributions

This paper has multiple authors and our individual contributions were as below

Statement (if applicable):

B.W. and X.W. conceived the article structure. B.W. and W.W. performed the rock mechanical experiments. J.B. developed energy balance criterion. B.W. numerical simulation. All the authors analysed the data. N.M. prepared the plots. H.L. language revision. All the authors prepared the initial draft and revised the paper.

An Investigation on the effect of high energy storage anchor on surrounding rock conditions

Bowen Wu ^{1,2}, Xiangyu Wang ^{1,2*}, Jianbiao Bai ², Wenda Wu ^{1,2},
Ningkang Meng ^{1,2}, Huasheng Lin ³

¹ School of Mines, China University of Mining & Technology, Xuzhou 221116, China

² State Key Laboratory of Coal Resources and Safe Mining, Xuzhou 221116, China

³ School of Minerals and Energy Resources Engineering, University of New South Wales, Sydney, NSW, 2052, Australia

Keywords: Energy evolution, tension bolts, UDEC, differential analysis, rock

1. Summary

High pre-tension bolt is an effective strata control technique and is the key to ensure the stability of anchorage and roadway. Based on the performances of high energy storage tension rock bolts in different rock properties, this study proposed a constitutive model to describe the energy balance of anchor under uniaxial compression. UDEC was used to simulate the behaviour of anchor in coal under uniaxial compression and results were analysed to study the rock mechanical properties, degree of damage and energy evolution. Simulation results showed that tension rock bolts can improve the mechanical properties and energy storage capacities of anchor. The energy evolution was divided into three stages: i) the external work was stored in the form of elastic strain energy (U^e) in the anchor prior to the yielding strength; ii) the elastic strain energy reached its maximum near the peak strength; iii) Energy was dissipated from fracture friction (W_f), plastic deformation (W_p) and acoustic emission (U^r) during post-peak stage. The installation of tension rock bolts resulted in the most improvement of surrounding rock condition for medium hard rock including sandy mudstone, whereas it had the least impact on hard rock, e.g. sandstone.

2. Introduction

Rock strength is a critical factor for underground structural design. To improve the mechanical properties of rock, rock and cable bolts are generally implemented. In underground coal mines, the rock strengthening measure is immediate bolt support after excavation to ensure the roadway stability and subsequent workplace safety [1]. Extensive practical work revealed that this is an effective strata control method, which has also been widely used in Chinese coal mines [2].

To determine the effect of rock bolt on anchor strengthening and improve the ability of anchorage, a significant body of researches have carried out. Hou [3] theoretically analysed the effect of rock bolts on the peak and the residual strengths of rock around anchorage range. Based on the improvement of mechanical properties prior and after the peak strength, Hou [3] proposed a constitutive model for rock strengthening effect around the roadway with rock bolts. Kang et al. [4] described the mechanisms and technologies of rock bolt supporting system in China. The authors suggested that high strength-stiffness-reliability and low-density proactive rock bolts can improve the strength of surrounding rock. Wang et al. [5] derived the equations of shear strength prior and after anchorage. The analysis showed that once the grouting is completed, the shear strength of anchorage range was improved together with the mechanical properties of surrounding rock (elastic modulus, cohesion, friction angle and dilation angle). Wei et al. [6] carried out physical simulation tests to study the influence of pre-tension on anchor strength and deformation and found that high pre-tension can improve the anchorage strength. Based on the triaxial physical simulation tests,

*Author for correspondence (wangxiangyu_cumt@163.com).

†Present address: Nanhu campus, China University of mining and technology, No.1, University Road, Xuzhou City, Jiangsu Province, China

Wang et al. [7] revealed that the rock bolt does not only improve the capacity of fracture surrounding rock, but also controls the discontinuous deformation of fractured surrounding rock. This will in turn improve the continuity and integrity of surrounding rock mass of deep roadway.

In numerical simulation, the bolt section in anchorage range can be represented by material model as well as structural elements. Both methods have been proven successful in simulating the mechanical properties of bolts and anchors under various loading conditions. Meng [8] imported the constitutive model of anchor in fractured rock to FLAC3D and subsequently simulated the effect of anchoring condition on ultimate bearing capacity. Results showed that the higher pre-tension and density can improve the ultimate bearing capacity. Zuo et al. [9] used elastic-isotropic model in FLAC3D to study the influence of the alignment between bolt and borehole dimeters on anchorage effect. Numerical simulation suggested a reasonable alignment (4 – 11 mm diameter difference) can improve the anchorage effect. D.-A. Ho et al. [10] used elasto-plastic model to study the fracture mechanism of interface between rock bolt and anchoring agent under various horizontal stress conditions. Thereby, Grasselli [11], Aziz and Jalalifar [12] and Tatone et al. [13] also used material model to simulate various laboratory condition of rock bolt. However, the mesh size is an important factor which can influence the simulation results using material model. This constrains the preciseness of the simulation. On the other hand, structural element approach can overcome this problem. At present, the structural element can be divided into 'cable element' and 'rockbolt element'. Zhang et al. [14] used cable element to study the parameters including length, anchorage length and rock anchor spacing via FLAC3D and proposed several suggestions to support bolt design. Wu et al. [15] attempted to simulate the supporting element (e.g. wooden cylinder) in supported roadway using Cable element and found that the rock bolts can improve the capacity of wooden cylinder and prevent the fracture development. Pull-test or underground excavation with rock bolt support were simulated by (Vardakos et al. [16], Malmgren and Nordlund [17], Li et al. [18], Gao et al. [19], Shreedharan and Kulatilake [20]). Ma et al. [21] used rockbolt element to simulate pull-test via FLAC2D and studied the relationship between rock bolt and rock as well as the mechanical properties of newly developed rock bolt and its preferable conditions.

According to the previous studies, it can be found that the focus was mainly on the anchorage technique, rock bolt, coordination effect and stress transfer. Although there is an overall agreement that high pre-tension can improve the mechanical properties of anchor, the influence of high pre-tension under various rock strength conditions on anchorage was not investigated. Majority of the research analysed the anchorage effect via elasto-plastic model and stress-strain relationship, while there were limited studies investigated the problem using energy conservation and the fracture evolution of anchorage is a process of energy dissipation and release. Hence, it is more reasonable to study the anchorage effect using energy law (Xie et al.[22]). Many numerical simulation use cable element to approach this topic; this is inappropriate considering the method is one-dimensional element which consists of two degrees of freedom. On the other hand, rockbolt element is more suitable for the analysis as it is a two-dimensional element with three degrees of freedom (Itasca Consulting Group Inc. [23]). The paper used rockbolt element through UDEC Trigon to study the influence of high pre-tension on anchor with various surrounding rock strengths under uniaxial compression. This aims to study the different strengthening behaviours of tension rock bolt under various conditions based on monitor of the energy evolution, the stress-strain relation as well as the fracture development in different rock strengths.

3. Energy balance and components

Nowadays, the study of energy evolution in rock mechanics has been refined to different depth of cover, loading conditions, in-situ stress magnitudes (Salamon [24], Napier [25], Onur Vardar et al. [26], Zhang et al. [27], Wang et al. [28]). In UDEC, the energy exchanged based on the work done on rock, joint and boundary. Without considering dynamic calculation, the energy balance in UDEC mainly involves work done on boundary, stored strain energy and dissipated energy (Itasca Consulting Group Inc. [23]). This study determined the constitutive law based on the previous studies as well as the simulation conditions.

In uniaxial compression, work done from external force is denoted by W , U^e is used to represent the elastic strain energy due to elastic deformation of the anchor. The difference in work done between boundary and elastic strain energy is dissipated energy, U^d . Hence, total input energy can be written as:

$$W = U^d + U^e \quad (1)$$

Energy is mainly dissipated in three ways. Firstly, energy is dissipated by friction due to the fracture development in rock; this is denoted as W_f . Second source is due to the plastic deformation of rock (W_p). Once the rock is undergone plastic deformation, energy dissipated from plastic work done. The rest of energy is dissipated via acoustic emission, denoted as U_r . Therefore, U^d can be expressed as:

$$U^d = W_f + W_p + U^r \quad (2)$$

In UDEC, the incremental change of these energy components is cumulative and determined by timesteps (Itasca Consulting Group Inc. [23]). By combining Eq (1) and Eq (2), W can be expressed as:

$$W = W_f + W_p + U^r + U^e \quad (3)$$

Subsequently, this energy balance equation can be used to investigate the complex energy dissipation process in anchor.

4. Parameters calibration

4.1. The UDEC Trigon approach

UDEC Trigon model was proposed by Gao and Stead [29], aiming at simulating brittle fracture of rock. In this model, a rock is represented by an assembly of triangular blocks bonded together via the grain contacts. Each block is made elastically by dividing them into triangular finite difference zones. In the direction normal to a contact, the stress-displacement relation is assumed to be linear and governed by the stiffness k_n , such that:

$$\Delta\sigma_n = -k_n \Delta u_n \quad (4)$$

where $\Delta\sigma_n$ is the effective normal stress increment and Δu_n is the normal displacement increment. A limiting tensile strength (T) is assumed for the contact. If this value is exceeded, then $\sigma_n = 0$.

In the shear direction, the response is governed by constant shear stiffness. The shear stress, τ_s , is determined by a combination of contact properties, cohesion (c) and friction angle (ϕ), where:

$$|\tau_s| \leq c + \sigma_n \tan\phi = \tau_{max} \quad (5)$$

then

$$\tau_s = -k_s \Delta u_s^e \quad (6)$$

however, if $|\tau_s| \geq \tau_{max}$, then:

$$\tau_s = \text{sign}(\Delta u_s^e) \tau_{max} \quad (7)$$

where Δu_s^e is the elastic component of the incremental shear displacement and Δu_s is the total incremental shear displacement.

The proposed modeling approach has been implemented in UDEC (Itasca Consulting Group Inc.[23]).

4.2 Mechanical parameters of coal and rock mass

Since blocks in UDEC Trigon model are considered as elastic materials, they cannot be plastically destroyed. However, the anchor breakage is plastic damage. To closely mimic the mechanical properties and energy evolution of anchor post peak strength, this study also used strain softening model in UDEC. The strain-softening model is based on the UDEC Mohr–Coulomb model with non-associated shear and associated tension flow rules.

This study selected coal (soft), sandy mudstone (medium) and sandstone (strong) to represent various rock strengths. The intact properties of the rock masses are listed in Table 1. These properties were obtained through laboratory compression tests and were provided by the Yuwu coal mine. The RQD values of the rock masses were evaluated from borehole televiewer images.

Table 1 Intact rock properties and scaled rock mass properties of coal measures from Yuwu coal mine

The rock mass elastic modulus was calculated using the relationship between RQD and the elastic modulus ratio (Zhang and Einstein [30]), as shown in Eq. 8, where E_m is the elastic modulus of the rock mass and E_r is the elastic modulus of the rock sample.

$$\frac{E_m}{E_r} = 10^{0.0186RQD-1.91} \quad (8)$$

The rock mass strength was then calculated using the relation between the unconfined compressive strength ratio σ_{cm} / σ_c and the deformation modulus ratio E_m / E_r (Singh and Seshagiri Rao [31]). The value of q is 0.63 (Gao. et al, [32]):

$$\frac{\sigma_{cm}}{\sigma_c} = \left(\frac{E_m}{E_r} \right)^q \quad (9)$$

To represent the coal and rock by using an assembly of triangular blocks, the properties of the blocks and contacts were calibrated against the coal measure and rock mass properties listed in Table 2 to 4. This was

achieved by simulating unconfined compression tests in a numerical model created using the Trigon logic. The size of the rock sample is 1 m (in width) × 2 m (in height) (Singh and Seshagiri Rao [31]) (Fig. 1). The bottom of the numerical model was fixed and a loading rate of 0.02 m/s was applied at the top. The calibrated properties of the UDEC model are illustrated in Table 5.

The uniaxial compressive strength and elastic modulus data derived by numerical simulation are similar to the data obtained from laboratory tests (within an error of 9%). Hence, the micromechanical parameters of the coal mass and the rock mass were properly calibrated.

Table 2 Calibrated mechanical parameters of blocks and joints of the coal measures

Table 3 Calibrated mechanical parameters of blocks and joints of sandy mudstone

Table 4 Calibrated mechanical parameters of blocks and joints of sandstone

Table 5 Calibrated microproperties in the UDEC Trigon model to represent the rock masses

Fig.1 Results of uniaxial compressive strength tests a numerical simulation of uniaxial compression, b stress–strain curve of coal, c stress–strain curve of sandy mudstone, d stress–strain curve of sandstone

3.3 Rockbolt element parameter

In UDEC, rockbolt element is different from cable element. Rockbolt element is a two-dimensional element which its two nodes have three degrees of freedom (two displacements and one rotation). It can resist bending and can yield along axial direction. Fig. 2 illustrates the components of rockbolt element. Rockbolts interact with UDEC via shear and normal coupling springs, which the springs are non-linear connectors. These are used to transfer force and motion between rockbolt element and the mesh nodes. Fig. 2 shows material behavior of shear and normal coupling spring for rockbolt elements. Shear coupling springs are used to simulate the shear behaviour of anchorage length and normal coupling spring are used to simulate the compression of surrounding rock.

Fig.2 Conceptual mechanical representation of the rockbolt element

Rockbolt element can be broken at the node. The bolt breakage can be simulated by defining the tensile failure strain limit (*tfstrain*). Rock bolt will be deemed as failed if $\varepsilon_{pl} \geq tfstrain$, where *tfstrain* has to be defined. ε_{pl} is the total tensile yield strain of any element of anchor:

$$\varepsilon_{pl} = \sum \varepsilon_{pl}^{ax} + \sum \frac{d}{2} \frac{\theta_{pl}}{L} \quad (10)$$

ε_{pl}^{ax} is the axial deformation; d is rock bolt diameter, θ_{pl} is the average rotation angle of the component.

This study used rockbolt element to simulate HRB335 threaded steel anchor at 22 mm diameter. Based on the laboratory pull-test results, the input parameters of rockbolt element can be obtained, as displayed in Table 6.

Table 6 Rockbolt and anchor parameters

5. Research background and model establishment

The common way to apply pre-tension is to install the nut on the tail thread of rock bolt. By applying torque to the nut, axial tension is acting onto the rock bolt (Li et al. [33]). However, high-pretension is difficult to be controlled in practice. For instance, when the torque achieves 300 – 400 Nm, the pre-tension can sometimes still be low. This low conversion efficiency is due to the torque at thread. To increase the pre-tension of rock bolt, a new type of anchor lock has been developed by the research team. This development changed the traditional pre-tension method, which is capable of applying high pre-tension without damaging rock bolt. The anchor lock is composed of an anchor ring and three clips. The anchor ring is a ring-shaped structure, in which the center hole is tapered. The clips are disposed between the inner wall of the tapered hole of the anchor ring and the anchor body, as displayed in Fig. 3.

Fig.3 Rockbolt barrel and wedge and assembly drawing

Fig. 4 shows the setup of UDEC Trigon simulation. The blocks and interfaces used strain softening and Coulomb-sliding models, respectively. The dimensions of the numerical model are 1 m × 2m. Due to the

limitation of rockbolt element, the effect of vertically installed rock bolts was not significant. Hence, rock bolts were installed horizontally to the rock body at $y = 0.4\text{m}$, 1.0m and 1.6m . Pre-tension was applied by applying load at the nodes of the rock bolts. In practice, the common torque applied to the rock bolt is approximately $300 - 400\text{ Nm}$, which are $40 - 50\text{ kN}$ pre-tension correspondingly (Kang et al. [34]). To achieve high pre-tension in the numerical model, the study set the pre-tension as 70% of the yielding strength, i.e. $138.6\text{ kN}/0.365\text{ MPa}$. The loading rate of 0.02 m/s was applied at the top boundary of the model while fixing the bottom boundary. During the simulation process, the stress-strain relationship and fracture development of anchor was monitored by FISH function. Boundary, frictional and plastic work were recorded using UDEC energy element. 'SET energy on' command was used to activate the energy monitoring function while turn off 'mass-scaling'. As this study used non-viscous boundary which does not consider the dynamic calculation, damping was set as 'auto'. The simulation procedures involved: i) model construction and calibration; ii) test simulation.

Table 7 Numerical simulation scheme

Fig.4 Model overview and dimensions

6. Discussion

6.1 Mechanical properties and damage analysis

To monitor the degree of damage of rock and anchor, the total fracture length as well as the shear and tensile fracture lengths during uniaxial compression were measured. The anchor and surrounding rock fracture are a complex process, which the shear and tensile failures can occur in a single fracture. Hence, only the initial (first) failure type was recorded for each fracture. The degree of damage (D) can be calculated as:

$$D = \frac{L_s + L_t}{L_c} \times 100\% \quad (11)$$

where L_c is the total fracture length, L_s and L_t are total shear and tensile fracture lengths, respectively.

Fig.5 The stress and degree of damage of rock with and without anchorage a coal without anchorage; b coal with anchorage; c sandy mudstone without anchorage; d sandy mudstone with anchorage; e sandstone without anchorage; f sandstone with anchorage

Fig. 5 illustrates the stress and degree of damage of rock during the process with and without anchorage. Based on the comparison, the mechanical properties (peak strength, residual strength and elastic modulus) of rock were increased with the installation of tension rock bolts, while the degree of damage decreased. Under the same axial strain, the higher strength always led to higher degree of damage regardless of the anchorage condition. The installation of tension rock bolts did not have significant influence on the axial strain of rock during damage. There was increase in the degree of damage prior to the peak rock strength and the gradient became flatter when it was near the peak strength. The indication here is that installation of tension bolts can prevent the fracture development and coalescence, such that the degree of damage was reduced. To compare the effect of tension rock bolts under different rock conditions, Table 8 summaries the critical parameters from Fig. 5.

Table 8 Mechanical properties and degree of damage of rock prior and after anchorage

Table 8 suggested that there are differences in the strengthening effect of tension rock bolts on rock mechanical properties with various rock types. When the bolt was installed in coal, elastic modulus, peak strength and residual strength increased 4.1% (0.03 GPa), 5.9% (0.3 MPa) and 5.2% (0.2 MPa), while the degree of damage decreased 13.7%. In sandy mudstone, elastic modulus, peak strength and residual strength increased 20.7% (0.6 GPa), 3.6% (0.9 MPa) and 13.1% (1.7MPa), while the degree of damage decreased 11.9%. For sandstone, elastic modulus, peak strength and residual strength increased 21.% (0.1 GPa), 0.6% (0.2 MPa) and 0.5% (0.1 MPa), while the degree of damage decreased 5.3%. Sandy mudstone had the most change in absolute magnitudes and sandstone had the least change. In terms of the relative magnitude difference, coal had the most change whereas sandstone had the least. The absolute magnitude change of sandy mudstone was the most while its degree of damage was only in the middle, indicating the mechanical properties of rock was improved by installation of rock bolts. However, this is not the primary reason for the reduction in degree of damage, in which the main reason was due to the prevention of fracture development and coalescence using rock bolts. Sandstone had the low change in terms of absolute and relative magnitudes. This suggests that high pre-tension rock bolts do not have significant influence on strong rock. This is because the rock strength was high and it required substantially higher stress to induce the rock failure. Under this condition, the effect of rock bolt is negligible comparing with the high strength. At post-peak of the loading process, the

fracture was more likely to be developed due to the low plasticity, which was difficult to be controlled using rock bolts. Overall, it can be concluded that tension rock bolts have insignificant influence on strong rock, such that the high pre-tension is not recommended for hard rock support design.

6.2 Analysis of the law of energy evolution

During the loading process, rock was continuously deforming and storing elastic strain energy. At the same time, the energy was dissipated due to internal damage. When the elastic strain energy reached the limit, it was released and led to rock failure. Hence, the failure process of rock is closely related to the energy evolution. Based on the analysis carried out in Section 5.1, tension rock bolts are not suitable for hard rock. Therefore, this section will focus on the energy evolution of coal and sandy mudstone, as displayed in Fig. 6 and 7.

Fig.6 Stress and energy evolution with and without anchorage in coal **a** stress-strain curve of coal (without anchorage); **b** a stress-strain curve of coal (without anchorage); **c** energy evolution of coal (without anchorage); **d** energy evolution of coal (anchorage)
 Fig.7 Stress and energy evolution with and without anchorage in sandy mudstone **a** stress-strain curve of sandy mudstone (without anchorage); **b** a stress-strain curve of sandy mudstone (without anchorage); **c** energy evolution of sandy mudstone (without anchorage); **d** energy evolution of sandy mudstone (anchorage)

In Fig. 6 and 7, the energy dissipation due to acoustic emission (U^r) was illustrated solely. This is because the energy dissipation curve of acoustic emission was similar as the accumulative acoustic emission, which is a major indication of anchor breakage. Secondly, the energy dissipation due to acoustic emission was substantially lower than the other sources. A sole illustration can represent the energy evolution of acoustic emission more clearly. According to the results from Fig. 6 and 7, the energy evolution of rock was consistent regardless of the rock bolts. Based on the correspondence between stress curve and energy curve, the energy evolution of rock can be divided into three stages:

Stage 1: From the start of the load to the yielding strength. External work was fully absorbed by rock and stored in the form of elastic strain energy. The energy curve increased non-linearly during this stage, while the external work curve was well aligned with the elastic strain curve. Based on the stress and energy curves, the rock was under elastic deformation prior to the yielding strength. There was no observable internal damage, such that the internal energy dissipation was considered as none.

Stage 2: From the yielding strength to the peak strength. External work continuously increased and the total energy that the rock absorbed was also increasing. However, as the rock was in plastic condition and there was some degree of internal damage, the frictional work in-between cracks and plastic deformation started to occur and increased at a high rate. This indicates that the degree of damage was decreased although the acoustic emission level was still low. During this stage, the elastic strain energy was also increasing but the growth rate was flatter. Thereby, its alignment with the external work was getting worse and reached its maximum near the peak strength point.

Stage 3: Post-peak stage. External work continuously increased but the elastic energy in the rock was released at a high rate. There was considerable increase in the energy dissipation due to frictional work, plastic deformation and acoustic emission. The acoustic mission energy curve was growing rapidly, suggesting that the degree of damage was high. The external work was mainly transferred to the energy require for rock fracture towards the end of the simulation and the elastic strain energy was not stored anymore. When the stress dropped to the residual strength, the elastic strain energy and energy dissipation curves became flatter, although the energy continuously dissipated.

Although the energy evolution of rock was consistent regardless of the existence of rock bolts, the degree of damage and other parameters changed noticeably with the installation of rock bolts. Firstly, there were increasing in stress and decreasing in strain at anchor damage with rock bolts. For coal, the stress increased from 4.4 MPa to 4.8 MPa and strain reduced from 0.57% to 0.5%. For sandy mudstone, stress increased from 18 MPa to 20.6 MPa and strain reduced from 0.6% to 0.49%. Secondly, the peak elastic energy increased from 29.2 kJ to 35 kJ and from 202.7 kJ to 240.7 kJ for coal and sandy mudstone, respectively. When strain reached 1.2%, the external work of coal and sandy mudstone increased 26.2 kJ and 33.7 kJ. For coal, the work done from fracture friction was 35.1 kJ without rock bolt, accounting for 50.1% of the total input energy; whereas the energy was 37.4 kJ and took account of 38.9% with rock bolt. On the other hand, for sandy mudstone, the work done from fracture friction was 230.3 kJ without rock bolt, accounting for 58.8% of the total input energy; whereas the energy was 158.7 kJ and took account of 37.2% with rock bolt. Based on aforementioned data, the installation of tension rock bolts can improve the strength of surrounding rock and its storage of elastic strain energy. At the same time, it can also prevent fracture development and postpone the elastic strain energy release at post-peak. Fig. 8 shows the comparison of each energy categories. From the figure, it can be seen that the performance of sandy mudstone was better than coal. Hence, the indication here is that the tension

rock bolts can improve the mechanical properties and energy storage capacity of rock, which agrees with the conclusion from Section 6.1. Thereby, the most suitable rock type for the application is medium hard rock, e.g. sandy mudstone.

Fig.8 The comparison between coal and sandy mudstone on different energy categories

Fig.9 The stress and energy evolution of sandstone with and without anchorage **a** stress-strain relationship curve of sandstone (without anchorage); **b** stress-strain relationship curve of sandstone (with anchorage); **c** energy evolution of sandy mudstone (without anchorage); **d** energy evolution of sandy mudstone (with anchorage)

Fig. 9 depicts the stress and energy evolution of sandstone with and without anchorage. According to the figure, it can be found that the relationships of sandstone follow the same trend as those of coal. The difference between sandstone and coal was that the changes of frictional work and plastic deformation of sandstone were insignificant with rock bolts, indicating that the installation of rock bolts does not increase the plasticity of sandstone nor prevent fracture development. The plastic deformation curve was always at low level while the frictional work curve was always at high level. This phenomenon again supports that the fracture development was the main reason of rock failure. This agrees with the conclusion from stress and damage analysis; thereby proved that tension rock bolts are not effective for hard rock roadway support, e.g. sandstone.

7 Conclusion

Based on the UDEC simulation on the strengthening effect of tension rock bolts in different rock properties, a number of parameters were studied, including mechanical properties, degree of damage and energy evolution. The following conclusions were drawn from the numerical results:

(1) Tension rock bolts can improve the mechanical properties and energy storage capacity of anchor. Subsequently, it can reduce the degree of damage of the anchor while increasing the elastic strain energy storage and plastic deformation work. By suppressing energy dissipation due to fracture friction and delaying the post-peak elastic strain release, the stability of anchor can be increased.

(2) Based on the stress-strain and energy dissipation-strain relationships during uniaxial compression, the energy evolution of anchor was divided into three stages. In the first stage, the external work was fully absorbed by rock and was stored as elastic deformation energy. In the second stage, the axial stress surpasses the rock yielding strength, which work was done from fracture friction and plastic deformation. At the same time, the growth rate of elastic strain energy was reduced and the elastic strain energy reached the maximum value near the peak strength point. In the third stage, the energy stored in the rock was rapidly released from elastic strain energy. The work from fracture friction, plastic deformation and acoustic emission increased substantially. At the end of the process, the energy of external work was mainly transferred to rock failure, where the energy was dissipated via fracture friction, plastic deformation and acoustic emission. Among them, work from fracture friction and plastic deformation was the major contributing factor for energy dissipation.

(3) Based on stress and energy evolution analysis on different rock types, it can be seen that the tension rock bolts were not effective for hard rock. The elastic modulus, peak strength and residual strength increased 2.1% (0.1 GPa), 0.6% (0.2 MPa) and 0.5% (0.1 MPa), respectively. The implementation could not prevent the fracture development within the rock mass, such that it was not recommended for hard rock supporting design.

(4) The absolute magnitudes of mechanical properties of sandy mudstone increased more than that of coal. However, the relative magnitudes and reduction of degree of damage was lower than coal. This indicates that the improvement of mechanical properties of rock due to rock bolts was the secondary factor for degree of damage reduction, whereas its effect on preventing fracture development and coalescence was the main contributing factor. For energy evolution, the stress at initial damage increased 14.4% (2.6 MPa) and 9% (0.4 MPa) for sandy mudstone and coal. In addition, the peak elastic strain energy increased 18.7% (38 kJ) and 9.6% (5.8 kJ) for sandy mudstone and coal. The work done due to fracture friction decreased from 58.8% to 37.2% of the total energy for sandy mudstone; and it decreased from 50.1% to 38.9% for coal. Therefore, tension rock bolts is more suitable for medium hard rock (e.g. sandy mudstone), followed by soft rock (e.g. coal).

Acknowledgments

The authors are grateful for the valuable comments from the editors and the reviewer, which has substantially improved the quality of our work.

Funding Statement

This work is supported by the Fundamental Research Funds for the Central Universities (2019BSCX05).

Data Accessibility

Our data are deposited at Dryad.

Review URL: <https://datadryad.org/stash/share/Agf-g6O9idvPjujUn9zqr8m79qBwHC6MF69F5n4fWk8>

doi:10.5061/dryad.cvdncjt1w

Competing Interests

We have no competing interests.

Authors' Contributions

B.W. and X.W. conceived the article structure. B.W. and W.W. performed the rock mechanical experiments. J.B. developed energy balance criterion. B.W. numerical simulation. All the authors analysed the data. N.M. prepared the plots. H.L. language revision. All the authors prepared the initial draft and revised the paper.

References

- Navid Bahrani, John Hadjigeorgiou 2017 Explicit reinforcement models for fully-grouted rebar rock bolts. *J Rock Mech Geotechl Eng.* **9**,267-280. (<https://doi.org/10.1016/j.jrmms.2004.06.003>)
- Wen ZJ, Xing ER, Shi SS et al. 2018 Overlying strata structural modeling and support applicability analysis for large mining-height stopes. *J. Loss Prevent Proc.* **57**,94–100. (<https://doi.org/10.1016/j.jlp.2018.11.006>)
- Hou CJ, Gou PF 2000 Study on the strengthening mechanism of surrounding rock strength with bolting support. *Chin J Rock Mec Eng.* **19**, 342-345.
- Kang HP, Jiang TM, Gao FQ. Effect of pretension stress to rock bolting. *J Coal Soc.* **2007**;32(7):680-685.
- Wang Q, Qin Q, Jiang B et al. 2019 Study and engineering application on the bolt-grouting reinforcement effect in underground engineering with fractured surrounding rock. *Tunn. Undergr. Space Technol.* **84**,237-247. (<https://doi.org/10.1016/j.tust.2018.11.028>)
- Wei SJ, Gou PF 2012 Analogy simulation test on strengthening effect for pretention of bolts on anchorage body. *J China Coal Soc.* **37**, 1987-1993.
- Wang WJ, Dong EY, Zhao ZW et al. 2020 Experimental study on mechanical properties of anchorage body and on anchorage mechanism. *J China Coal Soc.* **45**,82-89.
- Meng B. 2013 Study on bearing characteristic of soft and fractured roadway surrounding rock anchorage unit and its applications in engineering. Ph.D. thesis, China University of Mining and Technology.
- Zuo JP, Wen JH, Li YD et al. 2019 Investigation on the interaction mechanism and failure behavior between bolt and rock-like mass. *Tunn. Undergr. Space Technol.* **93**:14. (<https://doi.org/10.1016/j.tust.2019.103070>)
- D.-A. Ho, M. Bost, J.-P. Rajo 2019 Numerical study of the bolt-grout interface for fully grouted rockbolt under different confining conditions. *Int J Rock Mech Min Sci.* **119**,168–179. (<https://doi.org/10.1016/j.jrmms.2019.04.017>)
- Grasselli G. 2005 3D behaviour of bolted rock joints: experimental and numerical study. *Intel J Rock Mech Min Sci.* **42**,13-24. (<https://doi.org/10.1007/s00603-015-0885-9>)
- Ma S, Nemcik J, Aziz N 2014 Simulation of fully grouted rockbolts in underground roadways using FLAC2D. *Canadian Geotech J.* **51**,911-920. (<https://doi.org/10.1139/cgj-2018-0470>)
- Xie HP, Ju Y, Li LY et al. 2008 Energy mechanism of deformation and failure of rock masses based on energy dissipation and energy release principles. *Chin J Rock Mech Eng.* **27**,1729-1938c.
- Itasca Consulting Group Inc.2014 UDEC (Universal Distinct Element Code), Version 6.0. Itasca, Minneapolis.
- Salamon MDG. 1984 Energy considerations in rock mechanics: fundamental results. *J S Afr Inst Min Metall.* **84**,233-46.
- Napier JAL.1991 Energy changes in a rockmass containing multiple discontinuities. *J S Afr Inst Min Metall.* **91**,145-57.
- Onur Vardar, Zhang CG, Ismet Canbulat et al.2019 Numerical modelling of strength and energy release characteristics of pillar-scale coal mass. *J Rock Mec Geotech Eng.* **11**,935-943. (<https://doi.org/10.1016/j.jrmge.2019.04.003>)
- Zhang ZZ, Deng M, Bai JB et al. 2020 Strain energy evolution and conversion under triaxial unloading confining pressure tests due to gob-side entry retained. *Inteal J Rock Mech Min Sci.* **126**:104184. (doi:10.1016/j.jrmms.2019.104184)
- Wang M, Zheng D, Wang K, Li W. 2018 Strain energy analysis of floor heave in longwall gateroads. *R. Soc. open sci.* **5**: 180691. (<http://dx.doi.org/10.1098/rsos.180691>)
- Gao FQ, Stead D, Coggan J. 2014 Evaluation of coal longwall caving characteristics using an innovative udec trigon approach. *Comput Geotech.* **55**,448–460. (<https://doi.org/10.1016/j.compgeo.2013.09.020>)
- Zhang L, Einstein HH 2004 Using RQD to estimate the deformation modulus of rock masses. *Int J Rock Mech Min Sci.* **41**,337–341. ([https://doi.org/10.1016/S1365-1609\(03\)00100-X](https://doi.org/10.1016/S1365-1609(03)00100-X))
- Singh M, Rao KS. 2005 Empirical methods to estimate the strength of jointed rock masses. *Eng Geol.* **77**,127–137. (<https://doi.org/10.1016/j.enggeo.2004.09.001>)
- Gao FQ, Stead D. 2014 The application of a modified voronoi logic to brittle fracture modelling at the laboratory and field

scale. *Int J Rock Mech Min Sci.* **68**,1–14. (https://doi.org/10.1016/j.ijrmms.2014.02.003)

33 Li ZB, Zhang N, Han Cl et al. 2012 Relationship between pre-tightening force and tightening torque. *J China U Min Technol.* **41**,189-193.

34 Kang HP, Wang Y, Gao FQ. 2016 Mechanical performances and stress states of rock bolts under varying loading conditions. *Tunn. Undergr. Space Technol.* **52**,138-146. (https://doi.org/10.1016/j.tust.2015.12.005)

35 Wang, Xiangyu (2020), An Investigation on the effect of high energy storage anchor on surrounding rock conditions, v3, Dryad, Dataset, <https://doi.org/10.5061/dryad.cvdncjt1w>

Tables

Table 1 Intact rock properties and scaled rock mass properties of coal measures from Yuwu coal mine

Lithology	Intact rock		RQD	Rock mass		
	E_r (GPa)	$\bar{\sigma}_r$ (MPa)		E_m (GPa)	$\bar{\sigma}_{cm}$ (MPa)	$\bar{\sigma}_{tm}$ (MPa)
Coal	2.6	10.8	75%	0.79	5.1	0.51
Sandy mudstone	5.4	35.5	90%	3.1	25	2.5
Sandstone	9.7	53.5	88%	5.2	36.1	3.6

Table 2 Calibrated mechanical parameters of blocks and joints of the coal measures

	Density (kg/m ³)	Young's modulus (GPa)	Cohesion (MPa)	Friction (°)	Tension (MPa)
Block	1400	0.79	1.6 ($\epsilon_p=0$) 1.1 ($\epsilon_p=0.04$) 0.6 ($\epsilon_p=0.15$)	27	0.9
Joint	Normal stiffness (GPa)	Tangential stiffness (GPa)	Cohesion (MPa)	Friction (°)	Tension (MPa)
	113	45.2	1.3	18	0.4

Table 3 Calibrated mechanical parameters of blocks and joints of sandy mudstone

	Density (kg/m ³)	Young's modulus (GPa)	Cohesion (MPa)	Friction (°)	Tension (MPa)
Block	1800	3.1	8.0 ($\epsilon_p=0$) 5.0 ($\epsilon_p=0.06$) 2.0 ($\epsilon_p=0.15$)	27	2.5
Joint	Normal stiffness (GPa)	Tangential stiffness (GPa)	Cohesion (MPa)	Friction (°)	Tension (MPa)
	372	149	6.2	18	1.8

Table 4 Calibrated mechanical parameters of blocks and joints of sandstone

	Density (kg/m ³)	Young's modulus (GPa)	Cohesion (MPa)	Friction (°)	Tension (MPa)
Block	2550	5.2	13.0 ($\epsilon_p=0$) 7.0 ($\epsilon_p=0.04$) 4.0 ($\epsilon_p=0.10$)	30	2.5
Joint	Normal stiffness (GPa)	Tangential stiffness (GPa)	Cohesion (MPa)	Friction (°)	Tension (MPa)
	665	266	11.2	18	1.8

Table 5 Calibrated microproperties in the UDEC Trigon model to represent the rock masses

Lithology	Young's modulus(GPa)			Compressive strength(MPa)		
	Target	Calibrated	Error(%)	Target	Calibrated	Error(%)
Coal	0.79	0.73	7	5.1	5.1	0
Sandy mudstone	3.1	2.9	6	25	25.1	0.4
Sandstone	5.2	4.7	9	36.1	36.3	5

Table 6 Rockbolt and anchor parameters

	Cross-section area (m ²)	Elastic modulus (GPa)	Tensile yield strength (kN)	Second moment of area(m ⁴)	tfstrain		
Rockbolt	3.8×10^{-4}	200	198	1.2×10^{-8}	0.012		
	Exposed perimeter (m)	Cohesive strength of shear coupling spring (MPa)	Stiffness of shear coupling spring (GPa)	Frictional resistance of the shear coupling spring (°)	Cohesive strength of normal coupling spring (MPa)	Stiffness of normal coupling spring (GPa)	Frictional resistance of the normal coupling spring (°)
Anchor parameters	0.07	1	8	45	200	20	0

Table 7 Numerical simulation scheme

	Rockbolt existence	Pre-tension (load)	Loading rate
Coal (soft)	No	/	0.02m/s
	Yes	0.365MPa	0.02m/s
Sandy mudstone (medium)	No	/	0.02m/s
	Yes	0.365MPa	0.02m/s
Sandstone (strong)	No	/	0.02m/s
	Yes	0.365MPa	0.02m/s

Table 8 Mechanical properties and degree of damage of rock prior and after anchorage

	Rock bolt existence	Elastic modulus (GPa)	Peak strength (MPa)	Residual strength (MPa)	Degree of damage (%)
Coal (soft)	No	0.73	5.1	3.8	35.1
	Yes	0.76	5.4	4.0	30.3
Sandy mudstone (medium)	No	2.9	25.1	13	46.2
	Yes	3.5	26	14.7	40.7
Sandstone (strong)	No	4.7	36.3	20.2	50.8
	Yes	4.8	36.5	20.3	48.1

Figure and table captions

Table.1 Intact rock properties and scaled rock mass properties of coal measures from Yuwu coal mine

Table.2 Calibrated mechanical parameters of blocks and joints of the coal measures

Table.3 Calibrated mechanical parameters of blocks and joints of sandy mudstone

Table.4 Calibrated mechanical parameters of blocks and joints of sandstone

Table.5 Calibrated microproperties in the UDEC Trigon model to represent the rock masses

Table.6 Rockbolt and anchor parameters

Table.7 Numerical simulation scheme

Table.8 Mechanical properties and degree of damage of rock prior and after anchorage

Figure.1 Results of uniaxial compressive strength tests **a** numerical simulation of uniaxial compression, **b** stress - strain curve of coal, **c** stress - strain curve of sandy mudstone, **d** stress - strain curve of sandstone

Figure.2 Conceptual mechanical representation of the rockbolt element

Figure.3 Rockbolt barrel and wedge and assembly drawing

Figure.4 Model overview and dimensions

Figure.5 The stress and degree of damage of rock with and without anchorage **a** coal without anchorage; **b** coal with anchorage; **c** sandy mudstone without anchorage; **d** sandy mudstone with anchorage; **e** sandstone without anchorage; **f** sandstone with anchorage

Figure.6 Stress and energy evolution with and without anchorage in coal **a** stress-strain curve of coal (without anchorage); **b** a stress-strain curve of coal (without anchorage); **c** energy evolution of coal (without anchorage); **d** energy evolution of coal (anchorage)

Figure.7 Stress and energy evolution with and without anchorage in sandy mudstone **a** stress-strain curve of sandy mudstone (without anchorage); **b** a stress-strain curve of sandy mudstone (without anchorage); **c** energy evolution of sandy mudstone (without anchorage); **d** energy evolution of sandy mudstone (anchorage)

Figure.8 The comparison between coal and sandy mudstone on different energy categories

1
2
3 Figure.9 The stress and energy evolution of sandstone with and without anchorage **a** stress-strain relationship
4 curve of sandstone (without anchorage); **b** stress-strain relationship curve of sandstone (with anchorage);
5 **c** energy evolution of sandy mudstone (without anchorage); **d** energy evolution of sandy mudstone (with
6 anchorage)
7
8
9
10
11
12
13
14
15
16
17
18
19
20
21
22
23
24
25
26
27
28
29
30
31
32
33
34
35
36
37
38
39
40
41
42
43
44
45
46
47
48
49
50
51
52
53
54
55
56
57
58
59
60

Fig.1 Results of uniaxial compressive strength tests **a** numerical simulation of uniaxial compression, **b** stress–strain curve of coal, **c** stress–strain curve of sandy mudstone, **d** stress–strain curve of sandstone

Fig.2 Conceptual mechanical representation of the rockbolt element

Fig.3 Rockbolt barrel and wedge and assembly drawing

Fig.4 Model overview and dimensions

Fig.5 The stress and degree of damage of rock with and without anchorage **a** coal without anchorage; **b** coal with anchorage; **c** sandy mudstone without anchorage; **d** sandy mudstone with anchorage; **e** sandstone without anchorage; **f** sandstone with anchorage

Fig.6 Stress and energy evolution with and without anchorage in coal **a** stress-strain curve of coal (without anchorage); **b** a stress-strain curve of coal (without anchorage); **c** energy evolution of coal (without anchorage); **d** energy evolution of coal (anchorage)

Fig.7 Stress and energy evolution with and without anchorage in sandy mudstone **a** stress-strain curve of sandy mudstone (without anchorage); **b** a stress-strain curve of sandy mudstone (without anchorage); **c** energy evolution of sandy mudstone (without anchorage); **d** energy evolution of sandy mudstone (anchorage)

Fig.8 The comparison between coal and sandy mudstone on different energy categories

Fig.9 The stress and energy evolution of sandstone with and without anchorage **a** stress-strain relationship curve of sandstone (without anchorage); **b** stress-strain relationship curve of sandstone (with anchorage); **c** energy evolution of sandy mudstone (without anchorage); **d** energy evolution of sandy mudstone (with anchorage)

Table 1 Intact rock properties and scaled rock mass properties of coal measures from Yuwu coal mine

Lithology	Intact rock		RQD	Rock mass		
	E_r (GPa)	$\bar{\sigma}_r$ (MPa)		E_m (GPa)	$\bar{\sigma}_{cm}$ (MPa)	$\bar{\sigma}_m$ (MPa)
Coal	2.6	10.8	75%	0.79	5.1	0.51
Sandy mudstone	5.4	35.5	90%	3.1	25	2.5
Sandstone	9.7	53.5	88%	5.2	36.1	3.6

Table 2 Calibrated mechanical parameters of blocks and joints of the coal measures

	Density	Young's	Cohesion	Friction	Tension
	(kg/m ³)	modulus (GPa)	(MPa)	(°)	(MPa)
Block	1400	0.79	1.6 ($\epsilon_p=0$)	27	0.9
			1.1 ($\epsilon_p=0.04$)		
			0.6 ($\epsilon_p=0.15$)		
Joint	Normal	Tangential	Cohesion	Friction	Tension
	stiffness (GPa)	stiffness (GPa)	(MPa)	(°)	(MPa)
	113	45.2	1.3	18	0.4

Table 3 Calibrated mechanical parameters of blocks and joints of sandy mudstone

	Density	Young's	Cohesion	Friction	Tension
	(kg/m ³)	modulus (GPa)	(MPa)	(°)	(MPa)
Block	1800	3.1	8.0 ($\epsilon_p=0$)	27	2.5
			5.0 ($\epsilon_p=0.06$)		
			2.0 ($\epsilon_p=0.15$)		
Joint	Normal	Tangential	Cohesion	Friction	Tension
	stiffness (GPa)	stiffness (GPa)	(MPa)	(°)	(MPa)
	372	149	6.2	18	1.8

Table 4 Calibrated mechanical parameters of blocks and joints of sandstone

	Density	Young's	Cohesion	Friction	Tension
	(kg/m ³)	modulus (GPa)	(MPa)	(°)	(MPa)
Block	2550	5.2	13.0 ($\epsilon_p=0$)	30	2.5
			7.0 ($\epsilon_p=0.04$)		
			4.0 ($\epsilon_p=0.10$)		
Joint	Normal	Tangential	Cohesion	Friction	Tension
	stiffness (GPa)	stiffness (GPa)	(MPa)	(°)	(MPa)
	665	266	11.2	18	1.8

Table 5 Calibrated microproperties in the UDEC Trigon model to represent the rock masses

Lithology	Young's modulus(GPa)			Compressive strength(MPa)		
	Target	Calibrated	Error(%)	Target	Calibrated	Error(%)
Coal	0.79	0.73	7	5.1	5.1	0
Sandy mudstone	3.1	2.9	6	25	25.1	0.4
Sandstone	5.2	4.7	9	36.1	36.3	5

Table 6 Rockbolt and anchor parameters

	Cross-section area (m ²)	Elastic modulus (GPa)	Tensile yield strength (kN)	Second moment of area(m ⁴)	tfstrain		
Rockbolt	3.8×10 ⁻⁴	200	198	1.2×10 ⁻⁸	0.012		
	Exposed perimeter (m)	Cohesive strength of shear coupling spring (MPa)	Stiffness of shear coupling spring (GPa)	Frictional resistance of the shear coupling spring (°)	Cohesive strength of normal coupling spring (MPa)	Stiffness of normal coupling spring (GPa)	Frictional resistance of the normal coupling spring (°)
Anchor parameters	0.07	1	8	45	200	20	0

Table 7 Numerical simulation scheme

	Rockbolt existence	Pre-tension (load)	Loading rate
Coal (soft)	No	/	0.02m/s
	Yes	0.365MPa	0.02m/s
Sandy mudstone (medium)	No	/	0.02m/s
	Yes	0.365MPa	0.02m/s
Sandstone (strong)	No	/	0.02m/s
	Yes	0.365MPa	0.02m/s

Table 8 Mechanical properties and degree of damage of rock prior and after anchorage

	Rock bolt existence	Elastic modulus (GPa)	Peak strength (MPa)	Residual strength (MPa)	Degree of damage (%)
Coal (soft)	No	0.73	5.1	3.8	35.1
	Yes	0.76	5.4	4.0	30.3
Sandy mudstone (medium)	No	2.9	25.1	13	46.2
	Yes	3.5	26	14.7	40.7
Sandstone (strong)	No	4.7	36.3	20.2	50.8
	Yes	4.8	36.5	20.3	48.1

Appendix C

Response to Referees

The authors would like to appreciate the time and efforts of editor and reviewers on this manuscript. The authors have carefully considered all the points and suggestions and have incorporated major revisions into this manuscript. Revisions in the manuscript are marked in red and listed below are the authors' point-to-point responses to reviewers' comments:

Response to Reviewer#1 Comments

1 Page 3 line 50 'In UDEC, the energy exchanged based on the work done on rock, joint and boundary' is not clear.

The sentence has been revised for clearer understanding. This sentence comes from the UDEC user manual (udc521, Energy Calculation, pp 3-1). We have made some changes in the sentence pattern when quoting it, resulting in unclear expression. Now we replace this sentence with the original sentence 'Energy changes determined in UDEC are performed for the intact rock, the joints and for the work done on boundaries.'

2 Page 5 line 5-6 'The uniaxial compressive strength and elastic modulus data derived by numerical simulation are similar to the data obtained from laboratory tests (within an error of 9%).' Will be good to include this figure.

Figure 1 has been revised accordingly, including the laboratory test curves.

3 Why is the first paragraph of Chapter 5 listed separately? Is the development of new type of anchor lock a part of this paper?

The traditional way of bolt preload is applying torque, and it is difficult to apply high preload. Therefore, a lock for the bolt has been developed, which uses tension method to apply pre tension force to the rockbolt. This kind of rockbolt lock is relatively rare, so a paragraph was written to briefly introduce it. The lock and assembly mode are shown in Fig. 3 to let readers have a simple concept of this kind of lock

The development process of lock is not the research content of this paper, but the research of this paper is based on the rockbolt installed with lock. So a brief introduction is made

4 Page 6 line 40-41. 'The indication here is that installation of tension bolts can prevent the fracture development and coalescence, such that the degree of damage was reduced.' Not clear how you are saying this.

Fig. 5 illustrates the stress and degree of damage of rock during the process with and without anchorage. It can be seen from the figure that the mechanical properties (peak strength, residual strength, elastic modulus) of the rock mass before and after anchoring are not greatly improved, but the damage degree decreases obviously. In this paper, the damage degree is calculated based

on the crack length, the smaller the crack length, the smaller the damage degree. Therefore, the decrease of damage degree after the installation of tension rockbolt is mainly because the rockbolt inhibits the development of cracks. The follow-up discussion of the article also illustrates this point, but this conclusion is a bit abrupt here, so we deleted this sentence.

The degree of damage (D) can be calculated as:

$$D = \frac{L_s + L_T}{L_C} \times 100\%$$

where L_C is the total fracture length, L_s and L_T are total shear and tensile fracture lengths, respectively.

5 Page 7 line 1-2 ‘Overall, it can be concluded that tension rock bolts have insignificant influence on strong rock’ This conclusion is based on limited research, this expression is too absolute, it is suggest to rephrase.

The study of the whole paper shows that compared with the other two kinds of rock, the strengthening effect of tensile rockbolt on hard rock is relatively weak. However, the sentence pointed out by experts was too absolute at the beginning of the discussion, so we deleted it and described it in a relatively mild way in the conclusion.

6 The arrow mark in Figure 4 is not clear. The order of annotation in the title of Figure 6 is inconsistent with that in the figure.

The figures have been revised accordingly.

Response to Reviewer#2 Comments

1. 6.1 Mechanical properties and damage analysis. Table 8 presents the degree of damage for each rock type, with and without anchorage, at a particular strain level of 1.2%. It is not clear to the reader the reasons that the authors chose this level for damage degree comparison and not for example that one corresponds to the peak stress or any other strain level.

The greater the rock strength, the smaller the strain corresponding to damage. Therefore, when coal enters the post-peak stage, under the same strain, sandy mudstone and sandstone must also enter the post-peak stage. Therefore, using coal as the standard, select the appropriate strain level in the post-peak stage. The simulation found that when the strain is 1.2%, the growth of the coal damage curve tends to be stable, and the change after the strain 1.2% is small, so the strain 1.2% is selected

2. 6.2 Analysis of the law of energy evolution Page 7, lines 28-29: Fig. 5 shows that the degree of damage increasing in Stage 1 (before the yielding strength). Please comment and

clarify on this. Page 7, lines 34-35: This is in contrast to Fig. 5. Fig. 5 shows that the degree of damage is increasing at Stage 2. Please comment and clarify on this.

Thanks to the experts for pointing out the error, the increase in damage before the yield strength is indeed wrong. The reason for this error is that we used two FISH languages to monitor the damage, one of which did not work well. We uploaded this monitoring result by mistake.

The difference between these two FISH languages is the method of judging the threshold of crack generation. The wrong FISH language uses joint stiffness to determine whether cracks are generated, and the improved FISH language uses joint strength to determine. The stiffness of the joint changes with the increase of the normal stress [1] [2] [3]. Therefore, we want to judge whether the crack is generated by the change of stiffness. When the stiffness in the normal direction is greater than the normal stiffness, it is considered that a tensile crack is generated. If the stiffness in the shear direction is greater than the tangential stiffness, it is considered that a shear crack occurs. However, the disadvantage of this method is that the joint stiffness is too sensitive to changes in force (Figure 1) [3], which makes the monitoring results show that the damage increases before the yield strength. The strength determination method can avoid this error. The determination basis of this method is that when the force in the normal direction is greater than the tensile strength, it is determined as a tensile crack, and when the force in the shear direction is greater than the shear strength, it is determined that shear cracks were generated. Gao [4] used this method to correctly monitor the damage curve of the rock mass during uniaxial compression. We have listed the monitoring codes in the table below. The re-monitored damage curve has been uploaded, and the damage starts to rise near the yield strength, which is also consistent with the subsequent energy curve.

The description on Page 7, lines 34-35 is a translation error. We checked the original article. The original article describes the increase in damage in the second stage, not the decrease.

Figure.1 Relationship between crack stiffness and normal stress [3]

Fracture monitoring code		
call jmat.fin call contact.fin def set_ini ic = contact_head ; start of contact list loop while ic #0 ; loop through all blocks c_extra(ic) = 1 ic = c_next(ic) endloop end set_ini def _slipnum whilestepping ic=contact_head loop while ic # 0 if c_extra(ic) = 1 then cont_coh = fmem(c_jex(ic)+\$ac_coh)	cont_fri = mem(c_jex(ic)+\$ac_phi) cont_ten = fmem(c_jex(ic)+\$ac_ten) if c_length(ic) # 0 then nor_stress = c_nforce(ic)/c_length(ic) if nor_stress <= -0.98*cont_ten then c_extra(ic) = 2 endif shear_stress = abs(c_sforce(ic)/c_length(ic)) tmax = cont_coh + nor_stress*tan(cont_fri*3.14/180.0) if shear_stress >= tmax*0.98 then c_extra(ic) = 3 endif endif endif ic = c_next(ic) endloop	_tens = 0 _shears = 0 _add = 0 _crack = 0 ic = contact_head loop while ic # 0 _add = fmem(ic+\$KCL) if c_extra(ic) = 2 then _tens = _tens + _add endif if c_extra(ic) = 3 then _shears = _shears + _add endif _crack=_crack+c_length(ic) ic = c_next(ic) endloop _damage=(_tens +_shears)/_crack end hist_damage
The various data blocks, and the offsets of items within the blocks, are contained in a series of files supplied with UDEC. These files have the extension “.FIN” (for Fish INclude file); they provide symbolic names for offsets and current numerical values. The “.FIN” files serve two purposes: first, they document the meanings of the various data items; second, the files may be CALLed from a data file – they automatically execute and define appropriate symbols. The specific code of “contact. FIN” and “jmat. FIN” in pages 4-12 and 4-15 of the UDEC Manual UDC518.		

[1] Qiao Y, Hang C, Zhang L et al. 2019 Numerical simulation of fluid - solid coupling of fractured rock mass considering changes in fracture stiffness. Energy Science & Engineering. doi:10.1002/ese3.518

[2] Yao C, Jiang Q, Shao J, Zhou C. 2015 Modelling of hydro - mechanical coupling and transport in densely fractured rock mass. Eur J Environ Civ En. 19,521 - 538.

[3] Zhou J, Zhang LQ, Qi SW et al. 2020 Empirical ratio of dynamic to static stiffness for propped artificial fractures under variable normal stress. Eng Geo, 2020, 273.(<https://doi.org/10.1016/j.enggeo.2020.105683>)

[4] Gao FQ, Stead D. 2014 The application of a modified voronoi logic to brittle fracture modelling at the laboratory and field scale. Int J Rock Mech Min Sci. 68,1–14. (<https://doi.org/10.1016/j.ijrmms.2014.02.003>)

3. 7 Conclusions In this section there are some points that are mentioned extensively more than one time in Discussion section. For example, the point in page 8, lines 42-43 is

repeated in page 7, lines 1-2 and 9-10 as well as in page 8, lines 16-17. The authors should take care and reform properly the Discussion and the Conclusion section.

The authors have carefully considered the comment of the reviewer and revised the sections accordingly.

4. Is (are) there any Reference(s) for the values presented in Table 1?

The data in Table 1 comes from laboratory tests, mine geological data, borehole televiewer images, no other references.

5. If the bottom is fixed the model is wrong. Please comment and clarify. Use ‘displacement rate 0.02m/s’ instead of ‘loading rate 0.02m/s’

In UDEC, there are two loading methods for rock parameter calibration by uniaxial compression. One is fixed at the bottom and loaded at the top. Gao[1], Gao[2] and Wu[3] adopt this loading method. Another loading method is to apply loads at the top and bottom at the same time. Wu[4] adopts this loading method. Therefore, it is feasible to establish the model with fixed bottom. Fixed bottom refers to setting the bottom Y direction velocity to zero. These two loading methods have little influence on the simulation results, which can be eliminated by adjusting the loading speed.

Examples. Taking the sandy mudstone without anchorage as an example, the two loading modes are compared. In this paper, the bottom is fixed and the top loading speed is 0.02m/s. Now apply the load at the same time at the top and bottom of the model at a speed of 0.01m/s. The simulation curve comparison is shown in the figure below.

[1] Gao FQ, Stead D. 2014 The application of a modified voronoi logic to brittle fracture modelling at the laboratory and field scale. *Int J Rock Mech Min Sci.* 68,1–14. (<https://doi.org/10.1016/j.ijrmms.2014.02.003>)

[2] Gao F, Stead D, Kang H. 2015 Numerical simulation of squeezing failure in a coal mine roadway due to mining-induced stresses. *Rock Mech Rock Eng.* 48,1635- 1645. (<https://doi.org/10.1007/s00603-014-0653-2>)

[3] Wu BW, Wang XY, Bai JB et al. 2019 Study on crack evolution mechanism of roadside

backfill body in gob-side entry retaining based on UDEC Trigon model. *Rock Mech Rock Eng.* 52,3385–3399. (<https://doi.org/10.1007/s00603-019-01789-6>)

- [4] Wu WD, Bai JB, Wang XY et al. 2019. Numerical study of failure mechanisms and control techniques for a gob-side yield pillar in the Sijiazhuang coal mine, China. *Rock Mech Rock Eng* 52, 1231–1245. (<https://doi.org/10.1007/s00603-018-1654-3>)

‘Displacement rate 0.02m/s’ has been used in the article to replace ‘loading rate 0.02m/s’.

Appendix D

Response to Referees

The authors would like to appreciate the time and efforts of editor and reviewers on this manuscript. The authors have carefully considered all the points and suggestions and have incorporated revisions into this manuscript. Revisions in the manuscript are marked in red and listed below are the authors' point-to-point responses to reviewers' comments:

Response to Associate Editor Comments

1 Please correct "targe" in Table 5

Table 5 has been revised accordingly. "targe" has been corrected to "target".

2 Is "tension" in Tables 2,3,4 "tensile strength"?

Yes. The "tension" in Tables 2, 3, 4 has been changed to "tensile strength".

3 Rock mass does not have plural form. Please correct rock masses to rock mass

We have made corrections and marked red in the manuscript.

4 The sentence "The installation of tension rock bolts was more suitable for medium hard rock (e.g. sandy mudstone), whereas it was not effective for hard rock (e.g. sandstone)" is repeated in the summary text twice.

The repeated sentence has been deleted.

5 Fracture development of anchor was monitored by FISH function. -> why are you talking about fractures of the anchor?

The failure process of an anchor under load is not only the process of the generation, expansion, and penetration of fractures, but also the process of energy evolution. There is a certain correlation between them, and they are basically consistent at some key inflection points. In this article, monitoring the fracture length through FISH language defines the damage degree of the anchors, and assists in evaluating the supporting effects of different lithological anchors. Therefore, it is necessary to monitor fractures.

6 What is a Coulomb-sliding model. That is not a standard term.

In UDEC, the Coulomb slip model is a joint behavior model. The Coulomb slip models are described in Section 1.2.4 in Theory and Background (UDEC Manual).

We have corrected Coulomb-sliding model to Coulomb slip model. Coulomb slip model is a standard term.